# NETINFOF FRAMEWORK: MEASURING AND EXPLOITING NETWORK USABLE INFORMATION

**Meng-Chieh Lee**[1,*]**, Haiyang Yu**[2]**, Jian Zhang**[3]**, Vassilis N. Ioannidis**[3]**, Xiang Song**[3]**,
Soji Adeshina**[3]**, Da Zheng**[3,†]**, Christos Faloutsos**[1,3,†]

[1]Carnegie Mellon University, [2]Texas A&M University, [3]Amazon
{mengchil,christos}@cs.cmu.edu, {haiyang}@tamu.edu,
{jamezhan,ivasilei,xiangsx,adesojia,dzzhen}@amazon.com

## ABSTRACT

Given a node-attributed graph, and a graph task (link prediction or node classification), can we tell if a graph neural network (GNN) will perform well? More specifically, do the graph structure and the node features carry enough usable information for the task? Our goals are (1) to develop a fast tool to measure how much information is in the graph structure and in the node features, and (2) to exploit the information to solve the task, if there is enough. We propose NET-INFOF, a framework including NETINFOF_PROBE and NETINFOF_ACT, for the measurement and the exploitation of network usable information (NUI), respectively. Given a graph data, NETINFOF_PROBE measures NUI without any model training, and NETINFOF_ACT solves link prediction and node classification, while two modules share the same backbone. In summary, NETINFOF has following notable advantages: (a) *General*, handling both link prediction and node classification; (b) *Principled*, with theoretical guarantee and closed-form solution; (c) *Effective*, thanks to the proposed adjustment to node similarity; (d) *Scalable*, scaling linearly with the input size. In our carefully designed synthetic datasets, NETINFOF correctly identifies the ground truth of NUI and is the only method being robust to all graph scenarios. Applied on real-world datasets, NETINFOF wins in *11 out of 12* times on link prediction compared to general GNN baselines.

## 1 INTRODUCTION

Given a graph with node features, how to tell if a graph neural network (GNN) can perform well on graph tasks or not? How can we know what information (if any) is usable to the tasks, namely, link prediction and node classification? GNNs (Kipf & Welling, 2017; Hamilton et al., 2017; Veličković et al., 2018) are commonly adopted on graph data to generate low-dimensional representations that are versatile for performing different graph tasks. However, sometimes there are no network effects, and training a GNN will be a waste of computation time. That is to say, we want a measurement of how informative the graph structure and node features are for the task at hand, which we call *network usable information (NUI)*.

We propose NETINFOF, a framework to measure and exploit NUI in a given graph. First, NET-INFOF_PROBE measures NUI of the given graph with NETINFOF_SCORE (Eq. 2), which we proved is lower-bound the accuracy (Thm. 2). Next, our NETINFOF_ACT solves both the link prediction and node classification by sharing the same backbone with NETINFOF_PROBE. To save training effort, we propose to compute NETINFOF_SCORE by representing different components of the graph with carefully derived node embeddings. For link prediction, we propose the adjustment to node similarity with a closed-form formula to address the limitations when the embeddings are static. We demonstrate that our derived embeddings contain enough usable information, by showing the superior performance on both tasks. In Fig. 1, NETINFOF_ACT outperforms the GNN baselines most times on link prediction; in Fig. 2, NETINFOF_SCORE measured by NETINFOF_PROBE highly correlates to the test performance in real-world datasets.

---

*The work is done while being an intern at Amazon.
†Corresponding authors.

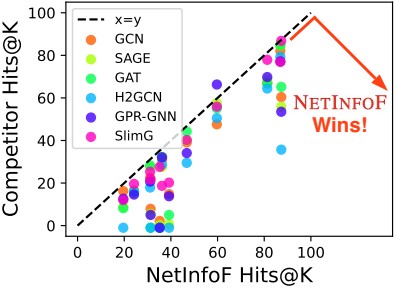

Figure 1: **NETINFOF wins** in real-world datasets on link prediction (most points are below or on line $x = y$).

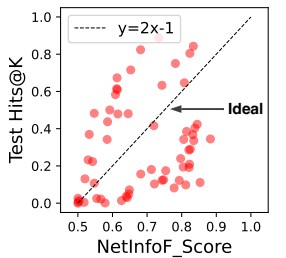
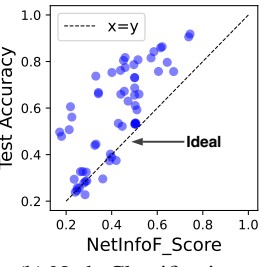

(a) Link Prediction     (b) Node Classification

Figure 2: **NETINFOF_SCORE highly correlates to test performance** in real-world datasets. Each point denotes the result of a component from each dataset.

In summary, our proposed NETINFOF has following advantages:

1. *General*, handling both node classification and link prediction (Lemma 1-2);
2. *Principled*, with theoretical guarantee (Thm. 1-2) and closed-form solution (Lemma 1-2);
3. *Effective*, thanks to the proposed adjustment of node similarity (Fig. 1);
4. *Scalable*, scaling linearly with the input size (Fig. 6).

In synthetic datasets, NETINFOF correctly identifies the ground truth of NUI and is the only method being robust to all possible graph scenarios; in real-world datasets, NETINFOF wins in *11 out of 12* times on link prediction compared to general GNN baselines.

**Reproducibility:** Code is at https://github.com/amazon-science/Network-Usable-Info-Framework.

## 2   RELATED WORKS

Table 1: **NETINFOF matches all properties**, while baselines miss more than one property.

| Property | | General GNNs | Subgraph GNNs | SlimG | NETINFOF |
|---|---|:---:|:---:|:---:|:---:|
| **1. General** w.r.t. Graph Task | 1.1. Node Classification | ✔ | | ✔ | ✔ |
| | 1.2. Link Prediction | ✔ | ✔ | ✔ | ✔ |
| **2. Principled** | 2.1. Theoretical Guarantee | | | | ✔ |
| | 2.2. Closed-Form Solution | | | ✔ | ✔ |
| **3. Scalable** | | ✔ | | ✔ | ✔ |
| **4. Robust** w.r.t. Input Scenario | 4.1. Node Classification | | | ✔ | ✔ |
| | 4.2. Link Prediction | | | | ✔ |

We introduce the related work in two groups: information theory, and GNNs. In a nutshell, NETINFOF is the only one fulfills all the properties as shown in Table 1.

**Information Theory.** The typical measure of the dependence between the random variables is the mutual information (Kraskov et al., 2004). It is powerful and widely used in sequential feature selection (Li et al., 2017), but its exact computation is difficult (Paninski, 2003; Belghazi et al., 2018) especially on continuous random variables (Ross, 2014; Mesner & Shalizi, 2020) and high-dimensional data (François et al., 2006; Mielniczuk & Teisseyre, 2019). Recently (Xu et al., 2020; Ethayarajh et al., 2022) proposed the concept of $\mathcal{V}$-information. However, the definition needs a trained model, which is expensive to obtain and is dependent on the quality of training.

Only a few works study the usable information in the graphs, but are not feasible in our problem settings because of three challenges, i.e., our desired method has to: (1) work without training any models, where Akhondzadeh et al. (2023) requires model training; (2) identify which components of the graph are usable, where Dong & Kluger (2023) ignores the individual components; and (3) generalize to different graph tasks, where Lee et al. (2022) focuses on node classification only.

**Graph Neural Networks.** Although most GNNs learn node embeddings assuming homophily, some GNNs (Abu-El-Haija et al., 2019; Zhu et al., 2020; Chien et al., 2021; Liu et al., 2021) break this assumption by handling $k$-step-away neighbors differently for every $k$, and some systematically study the heterophily graphs on node classification (Platonov et al., 2022; Luan et al., 2022; 2023; Mao et al., 2023; Chen et al., 2023; Ma et al., 2021). Subgraph GNNs (Zhang & Chen, 2018; Yin et al., 2022) are designed only for link prediction and are expensive in inference. On the other hand, linear GNNs (Wu et al., 2019; Wang et al., 2021; Zhu & Koniusz, 2021; Li et al., 2022; Yoo et al., 2023) target interpretable models. Such approaches remove the non-linear functions and maintain good performance. As the only method being robust to all graph scenarios, SLIMG (Yoo et al., 2023) works well on node classification. However, it is unclear how well it works for link prediction.

In conclusion, the proposed NETINFOF is the only one that fulfills all the properties in Table 1.

# 3 NETINFO_SCORE: WOULD A GNN WORK?

How to tell whether a GNN will perform well on the given graph task? A graph data is composed of more than one component, such as graph structure and node features. In this section, we define our problem, and answer two important questions: (1) How to measure the predictive information of each component in the graph? (2) How to connect the graph information with the performance metric on the task? We identify that a GNN is able to perform well on the task when its propagated representation is more informative than graph structure or node features.

## 3.1 PROBLEM DEFINITION

Given an undirected graph $G = (\mathcal{V}, \mathcal{E})$ with node features $\mathbf{X}_{|\mathcal{V}| \times f}$, where $f$ is the number of features, the problem is defined as follows:

- **Measure** the network usable information (NUI), and
- **Exploit** NUI, if there is enough, to solve the graph task.

We consider two most common graph tasks, namely link prediction and node classification. In link prediction, $\mathcal{E}$ is split into $\mathcal{E}_{\text{train}}$ and $\mathcal{E}_{\text{pos}}$. The negative edge set $\mathcal{E}_{\text{neg}}$ is randomly sampled with the same size of $\mathcal{E}_{\text{pos}}$. The goal is to predict the existence of the edges, 1 for the edges in $\mathcal{E}_{\text{pos}}$, and 0 for the ones in $\mathcal{E}_{\text{neg}}$. In node classification, $|\mathcal{V}_{\text{train}}|$ node labels $\mathbf{y} \in \{1, ..., c\}^{|\mathcal{V}_{\text{train}}|}$ are given, where $c$ is the number of classes. The goal is to predict the rest $|\mathcal{V}| - |\mathcal{V}_{\text{train}}|$ unlabeled nodes' classes.

## 3.2 PROPOSED DERIVED NODE EMBEDDINGS

To tell whether a GNN will perform well, we can analyze its node embeddings, but they are only available after training. For this reason, we propose to analyze the derived node embeddings in linear GNNs. More specifically, we derive 5 different components of node embeddings that can represent the information of graph structure, node features, and features propagated through structure.

***C1: Structure Embedding.*** The structure embedding $\mathbf{U}$ is the left singular vector of the adjacency matrix $\mathbf{A}$, which is extracted by the singular value decomposition (SVD). This aims to capture the community information of the graph.

***C2: Neighborhood Embedding.*** The neighborhood embedding $\mathbf{R}$ aims to capture the local higher-order neighborhood information of nodes. By mimicking Personalized PageRank (PPR), we construct a random walk matrix $\mathbf{A}_{\text{PPR}}$, where each element is the number of times that a node visits another node in $T$ trials of the $k_{\text{PPR}}$-step random walks. By doing random walks, the local higher-order structures will be highlighted among the entire graph. To make $\mathbf{A}_{\text{PPR}}$ sparser and to speed up the embedding extraction, we eliminate the noisy elements with only one visited time. We extract the left singular vectors of $\mathbf{A}_{\text{PPR}}$ by SVD as the neighborhood embeddings $\mathbf{R}$.

***C3: Feature Embedding.*** Given the raw node features $\mathbf{X}$, we represent the feature embedding with the preprocessed node features $\mathbf{F} = g(\mathbf{X})$, where $g$ is the preprocessed function.

***C4: Propagation Embedding without Self-loop.*** We row-normalize the adjacency matrix into $\mathbf{A}_{\text{row}} = \mathbf{D}^{-1}\mathbf{A}$, where $\mathbf{D}$ is the diagonal degree matrix. The features are propagated without self-loop to capture the information of $k_{\text{row}}$-step neighbors, where $k_{\text{row}}$ is an even number. This is useful to capture the information of similar neighbors when the structure exhibits heterophily (e.g., in a bipartite graph). Therefore, we have node embedding $\mathbf{P} = g(l(\mathbf{A}_{\text{row}}^2 \mathbf{X}))$, where $l$ is the column-wise L2-normalization, ensuring every dimension has a similar scale.

***C5: Propagation Embedding with Self-loop.*** The adjacency matrix with self-loop has been found useful to propagate the features in graphs that exhibit homophily. Following the most common strategy, we symmetrically normalize the adjacency matrix into $\tilde{\mathbf{A}}_{\text{sym}} = (\mathbf{D} + \mathbf{I})^{-\frac{1}{2}}(\mathbf{A} + \mathbf{I})(\mathbf{D} + \mathbf{I})^{-\frac{1}{2}}$, where $\mathbf{I}$ is the identity matrix. Similar to C4, we have node embeddings $\mathbf{S} = g(l(\tilde{\mathbf{A}}_{\text{sym}}^{k_{\text{sym}}} \mathbf{X}))$.

While C1-2 aim to capture the information with only the graph structure, C4-5 aim to capture the information of propagation, which is similar to the one that a trained GNN can capture. To ensure that the embeddings have intuitive meanings, we set all the number of steps $k_{\text{PPR}}$, $k_{\text{row}}$ and $k_{\text{sym}}$ as 2, which works sufficiently well in most cases. As C1-2 adopted SVD as their last step, the embedding

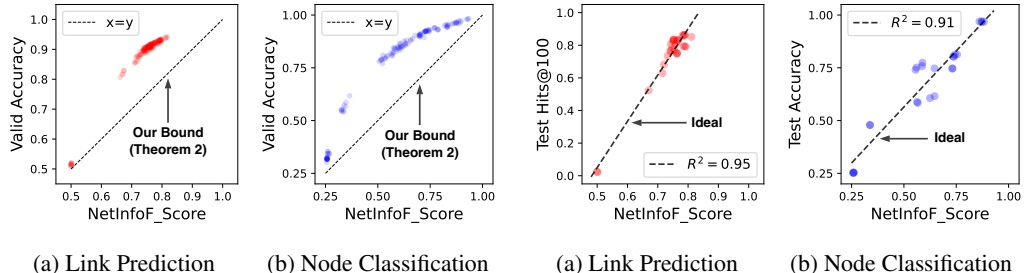

Figure 3: **Thm. 2 holds**. NETINFOF_SCORE is always less than or equal to validation accuracy. Figure 4: **NETINFOF_SCORE predicts right.** It is correlated to test performance in synthetic datasets.

dimensions are orthogonalized. For C3-5, we use principal component analysis (PCA) as $g$ to reduce and orthogonalize the embedding dimensions, leading to faster convergence and better performance when training the model. Each component has the same number of dimensions $d$.

### 3.3 NETINFOF_SCORE: DEFINITION AND THEOREMS

Next, we want to find a formula that connects the metrics of graph information and task performance. To begin, we derive the inequality between entropy and accuracy:

**Theorem 1** (Entropy and Accuracy). *Given a discrete random variable $Y$, we have:*

$$2^{-H(Y)} \leq accuracy(Y) = \max_{y \in Y} p_y \tag{1}$$

*where $H(Y) = -\sum_{y \in Y} p_y \log p_y$ denotes the Shannon entropy.*

*Proof.* See Appx. A.1. ∎

Before extending Thm. 1 to the case with two random variables, we need a definition:

**Definition 1** (NETINFOF_SCORE of Y given X). *Given two discrete random variables $X$ and $Y$, NETINFOF_SCORE of $Y$ given $X$ is defined as:*

$$\text{NETINFOF\_SCORE} = 2^{-H(Y|X)} \tag{2}$$

*where $H(\cdot|\cdot)$ denotes the conditional entropy.*

We prove that NETINFOF_SCORE low-bounds the accuracy:

**Theorem 2** (NETINFOF_SCORE). *Given two discrete random variables $X$ and $Y$, NETINFOF_SCORE of $Y$ given $X$ low-bounds the accuracy:*

$$\text{NETINFOF\_SCORE} = 2^{-H(Y|X)} \leq accuracy(Y|X) = \sum_{x \in X} \max_{y \in Y} p_{x,y} \tag{3}$$

*where $p_{x,y}$ is the joint probability of $x$ and $y$.*

*Proof.* See Appx. A.2. ∎

Thm. 2 provides an advantage to NETINFOF_SCORE by giving it an intuitive interpretation, which is the lower-bound of the accuracy. When there is little usable information to the task, the value of NETINFOF_SCORE is close to random guessing. To empirically verify it, we run the experiments on the synthetic datasets (Appx. D.1) with five splits, and report NETINFOF_SCORE and accuracy for all components of the derived embeddings. In Fig. 3, we find that even for the validation set, NETINFOF_SCORE is always less than or equal to the accuracy, strictly following Thm. 2. In the next sections, we show how NETINFOF_SCORE can be effectively and efficiently computed with our proposed NETINFOF_PROBE.

## 4 NETINFOF FOR LINK PREDICTION

With the derived node embeddings, how can we measure NUI in link prediction as well as solve the task? Compared to general GNNs, the node embeddings of linear GNNs are given by closed-form formula. They are thus rarely applied on link prediction because of following two reasons: (1) Predicting links by GNNs relies on measuring node similarity, which is incorrect if the neighbors are expected to have dissimilar embeddings; for example, in a bipartite graph, while a source node is connected to a target node, their structural embeddings are expected to be very different, resulting in low node similarity by linear GNNs; (2) In order to perform well on Hits@$K$, it is crucial to suppress the similarity of the nodes of negative edges, i.e. the unexisting connections in the graph. Hits@$K$ is the ratio of positive edges that are ranked at $K$-th place or above among both the positive and negative edges, which is preferred in link prediction where most real-world applications are recommendations. Since the embeddings of linear GNNs are static, they can not learn to separate the embeddings of nodes on each side of the negative edges. Therefore, how to generalize linear GNNs to solve link prediction remains a challenge.

For these reasons, we propose an adjustment to the similarity of the nodes, which generalizes NET-INFOF to link prediction, including NETINFOF_PROBE to measure NUI and NETINFOF_ACT to solve the task.

### 4.1 PROPOSED ADJUSTMENT TO NODE SIMILARITY

To solve the limitations of linear GNNs on link prediction, it is crucial to properly measure the similarity between nodes. We consider cosine similarity as the measurement, whose value is normalized between $0$ and $1$. By L2-normalizing each node embedding $\mathbf{z}_{1 \times d}$, the cosine similarity reduces to a simple dot product $\mathbf{z}_i \cdot \mathbf{z}_j$. However, even if node $i$ and node $j$ are connected by an edge, it may result in low value if they are expected to have dissimilar embeddings (e.g. structure embeddings in a bipartite graph). Therefore, before the dot product, we propose using the compatibility matrix $\mathbf{H}_{d \times d}$ to transform one of the embeddings, and rewrite the node similarity function into $\mathbf{z}_i \mathbf{H} \mathbf{z}_j^\mathsf{T}$.

The compatibility matrix $\mathbf{H}$ represents the characteristics of the graph: if the graph exhibits homophily, $\mathbf{H}$ is nearly diagonal; if it exhibits heterophily, $\mathbf{H}$ is off-diagonal. It is commonly assumed, given in belief propagation (BP) to handle the interrelations between node classes. In our case, $\mathbf{H}$ represents the interrelations between the dimensions of the node embeddings. By maximizing the similarity of nodes connected by edges, $\mathbf{H}$ can be estimated by the following lemma:

**Lemma 1** (Compatibility Matrix). *The compatibility matrix $\mathbf{H}$ has the closed-form solution and can be solved by the following optimization problem:*

$$\min_{\mathbf{H}} \sum_{(i,j) \in \mathcal{E}} \|\mathbf{z}_i \mathbf{H} - \mathbf{z}_j\|_2^2, \tag{4}$$

*where $\mathcal{E}$ denotes the set of (positive) edges in the given graph.*

*Proof.* See Appx. A.3. ∎

This optimization problem can be efficiently solved by multi-target linear regression. Nevertheless, this estimation of $\mathbf{H}$ does not take into account negative edges, which may accidentally increase the similarity of negative edges in some complicated cases. This hurts the performance especially when evaluating with Hits@$K$. Therefore, based on Lemma 1, we propose an improved estimation $\mathbf{H}^*$, which further minimizes the similarity of nodes connected by the sampled negative edges:

**Lemma 2** (Compatibility Matrix with Negative Edges). *The compatibility matrix with negative edges $\mathbf{H}^*$ has the closed-form solution and can be solved by the following optimization problem:*

$$\min_{\mathbf{H}^*} \sum_{(i,j) \in \mathcal{E}} \left(1 - \mathbf{z}_i \mathbf{H}^* \mathbf{z}_j^\mathsf{T}\right) - \sum_{(i,j) \in \mathcal{E}_{neg}} \left(\mathbf{z}_i \mathbf{H}^* \mathbf{z}_j^\mathsf{T}\right), \tag{5}$$

*where $\mathcal{E}_{neg}$ denotes the set of negative edges.*

*Proof.* See Appx. A.4. ∎

With great power comes great responsibility, estimating $\mathbf{H}^*$ has a higher computational cost than estimating $\mathbf{H}$. Thus, we provide three techniques to speed up the computation of $\mathbf{H}^*$ with the help of $\mathbf{H}$, and the details are in Algo. 1.

***T1: Warm Start.*** We approximate the solution by LSQR iteratively and warm up the approximation process with $\mathbf{H}$. Since $\mathbf{H}$ is similar to $\mathbf{H}^*$ and cheap to compute, this step largely speeds up the approximation process and reduces the number of iterations needed for convergence.

***T2: Coefficient Selection.*** We reduce the number of coefficients by only estimating the upper triangle of $\mathbf{H}^*$, and keep the ones with $95\%$ energy in $\mathbf{H}$. This is because the similarity function is symmetric, and the unimportant coefficients with small absolute values in $\mathbf{H}$ remain unimportant in $\mathbf{H}^*$. The absolute sum of the kept coefficients divided by $\sum_{i=1}^{d} \sum_{j=i+1}^{d} |\mathbf{H}_{ij}|$ is $95\%$ and the rest are zeroed out. This helps us reduce the number of coefficients from $d^2$ to be less than $(d+1)d/2$.

***T3: Edge Reduction.*** We sample $S$ positive edges from the 2-core graph, and $2S$ negative edges, where the sample size $S$ depends on $d$. Since in large graphs $|\mathcal{E}|$ is usually much larger than $d^2$, it is not necessary to estimate fewer than $(d+1)d/2$ coefficients with all $|\mathcal{E}|$ edges. Moreover, the 2-core graph remains the edges with stronger connections, where each node in it has at least degree 2. Sampling from the 2-core graph avoids interference from noisy edges and leads to better estimation.

## 4.2 NETINFOF_PROBE FOR NUI MEASUREMENT

Based on Thm. 2, we propose NETINFOF_PROBE that computes NETINFOF_SCORE, without exactly computing the conditional entropy of the high-dimensional variables. By sampling negative edges, the link prediction can be seen as a binary classification problem. For each component of embeddings, NETINFOF_PROBE esitmates its corresponding $\mathbf{H}^*$ and discretizes the adjusted node similarity of positive and negative edges. To avoid overfitting, we fit the $k$-bins discretizer with the similarity of training edges, and discretize the one of validation edges into $k$ bins. NETINFOF_SCORE can then be easily computed between two categorical variables. For instance, the node similarity between node $i$ and $j$ with embedding $\mathbf{U}$ is $(\hat{\mathbf{U}}_i \mathbf{H}_{\hat{\mathbf{U}}}^*) \cdot \hat{\mathbf{U}}_j$, where $\hat{}$ denotes the embedding preprocessed by column-wise standardization and row-wise L2-normalization. The details are in Algo. 2.

## 4.3 NETINFOF_ACT FOR NUI EXPLOITATION

To solve link prediction, NETINFOF_ACT shares the same derived node embeddings with NETINFOF_PROBE, and uses a link predictor following by the Hadamard product of the embeddings. We transform the embeddings on one side of the edge with $\mathbf{H}^*$, which handles the heterophily embeddings and better separates the nodes in the negative edges. By concatenating all components, the input to the predictor is as follows:

$$\underbrace{\hat{\mathbf{U}}_i \mathbf{H}_{\hat{\mathbf{U}}}^* \odot \hat{\mathbf{U}}_j}_{\text{Structure}} \| \underbrace{\hat{\mathbf{R}}_i \mathbf{H}_{\hat{\mathbf{R}}}^* \odot \hat{\mathbf{R}}_j}_{\text{PPR}} \| \underbrace{\hat{\mathbf{F}}_i \mathbf{H}_{\hat{\mathbf{F}}}^* \odot \hat{\mathbf{F}}_j}_{\text{Features}} \| \underbrace{\hat{\boldsymbol{P}}_i \mathbf{H}_{\hat{\boldsymbol{P}}}^* \odot \hat{\boldsymbol{P}}_j}_{\substack{\text{Features of} \\ \text{2-Step Neighbors}}} \| \underbrace{\hat{\mathbf{S}}_i \mathbf{H}_{\hat{\mathbf{S}}}^* \odot \hat{\mathbf{S}}_j}_{\substack{\text{Features of} \\ \text{Grand Neighbors}}} \tag{6}$$

where $(i, j) \in \mathcal{E} \cup \mathcal{E}_{\text{neg}}$. Among all the choices, we use LogitReg as the predictor for its scalability and interpretability. We suppress the weights of useless components, if there is any, by adopting sparse-group LASSO for the feature selection. The time complexity of NETINFOF_ACT is:

**Lemma 3.** *The time complexity of* NETINFOF_ACT *for link prediction is linear on the input size* $|\mathcal{E}|$:

$$O(f^2|\mathcal{V}| + f^3 + d^4|\mathcal{E}|) \tag{7}$$

*where $f$ and $d$ are the number of features and embedding dimensions, respectively.*

*Proof.* See Appx. A.5. ∎

## 5 NETINFOF FOR NODE CLASSIFICATION

In this section, we show how we can generalize NETINFOF to node classification. In contrast to link prediction, node classification does not rely on the node similarity, needing no compatibility matrix.

## 5.1 NETINFOF_PROBE FOR NUI MEASUREMENT

To effectively and efficiently compute NETINFOF_SCORE, we propose to assign labels to the nodes by clustering. This idea is based on the intuition that good embeddings for node classification can be easily split by clustering. Among clustering methods, we use $k$-means as it is fast. We cluster each component of the embeddings and compute NETINFOF_SCORE, where $k \geq c$. To ensure that the clustering is done stably, a row-wise L2-normalization is done on the embedding. The details are in Algo. 3.

## 5.2 NETINFOF_ACT FOR NUI EXPLOITATION

To solve node classification, we again concatenate the embeddings of different components, and the input of classifier is as follows:

$$ \underbrace{l(\mathbf{U})}_{\text{Structure}} \| \underbrace{l(\mathbf{R})}_{\text{PPR}} \| \underbrace{l(\mathbf{F})}_{\text{Features}} \| \underbrace{l(\mathbf{P})}_{\substack{\text{Features of} \\ \text{2-Step Neighbors}}} \| \underbrace{l(\mathbf{S})}_{\substack{\text{Features of} \\ \text{Grand Neighbors}}} \tag{8} $$

where $l$ is the column-wise L2-normalization. Similar to NETINFOF_ACT in link prediction, we use LogitReg as the classifier and adopt sparse-group LASSO for the regularization.

## 6 SYNTHETIC DATASETS FOR SANITY CHECKS

To ensure that NETINFOF is robust to all graph scenarios, we carefully design the synthetic datasets for sanity checks. We include all possible graph scenarios, where the ground truth of NUI is available. The details of implementation are in Appx. D.1.

### 6.1 LINK PREDICTION

**Designs.** We separate the nodes into $c$ groups to simulate that there are usually multiple communities in a graph. To cover all the possibilities in the real-world, the scenarios are the cross-product of different scenarios on the node features $\mathbf{X}$ and the graph structure $\mathbf{A}$, as shown in Fig. 5. We ignore the scenario that $\mathbf{X}$ is useful but $\mathbf{A}$ is useless, since this is impractical in the real-world.

There are 3 scenarios of node features $\mathbf{X}$:

1. **Random**: the node features are random, with no correlation with the existence of edges.
2. **Global**: all dimensions of the node features are correlated with the existence of edges.
3. **Local**: only a subset of dimensions of the node features are correlated with the existence of edges, where there is no overlapping between the subsets of node groups.

There are 2 scenarios of graph structure $\mathbf{A}$:

1. **Diagonal**: the nodes and their neighbors are in the same group.
2. **Off-Diagonal**: the nodes and their neighbors are in two different groups.

**Observations.** In Table 2, NETINFOF receives the highest average rank among all GNN baselines, and is the only method that can handle all scenarios. While GNNs have worse performance when $\mathbf{X}$ is either random or local, SLIMG, a linear GNN, cannot handle cases with off-diagonal $\mathbf{A}$.

**Would a GNN work?** Fig. 4a shows that NETINFOF_SCORE is highly correlated with test Hits@100, with high $R^2$ values, where each point denotes a component of embeddings from each split of synthetic datasets. In Appx. C.1, Table 6 reports NETINFOF_SCORE and test performance of each component. By measuring their NETINFOF_SCORE, NETINFOF_PROBE tells when propagating features through structure contains less information than using features or structure itself. For example, in scenarios where node features are useless (the first two scenarios in Table 6), NET-INFOF_PROBE spots that $\mathbf{F}$ (i.e., $g(\mathbf{X})$) provides little NUI to the task, and thus the propagated embeddings $\mathbf{P}$ and $\mathbf{S}$ have less NUI than the structural embeddings $\mathbf{U}$ and $\mathbf{R}$. This indicates that training GNNs is less likely to have a better performance than only utilizing the information from the graph structure, which correctly matches the test performance in Table 6.

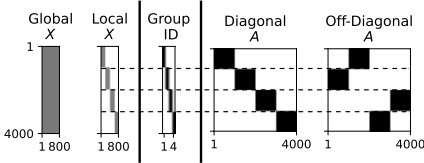

Figure 5: Scenarios of node features $\mathbf{X}$ and graph structure $\mathbf{A}$ in synthetic datasets.

Table 2: **NETINFOF wins** on link prediction in the synthetic datasets. Hits@100 is reported.

| Model | Rand. X Diag. A | Rand. X Off-Diag. A | Global X Diag. A | Global X Off-Diag. A | Local X Diag. A | Local X Off-Diag. | Avg. Rank |
|---|---|---|---|---|---|---|---|
| GCN | 82.7±1.1 | 70.9±1.2 | 87.2±0.5 | 85.1±1.2 | 17.4±1.7 | 19.2±1.8 | 4.3 (0.9) |
| SAGE | 77.4±1.1 | 66.5±4.1 | 86.2±1.0 | 85.2±2.2 | 11.0±1.2 | 09.5±1.0 | 5.1 (1.1) |
| GAT | 86.3±0.9 | 83.1±0.3 | 87.3±0.9 | 85.2±0.6 | 16.0±2.0 | 16.9±2.1 | 3.3 (1.5) |
| H²GCN | 24.5±4.3 | 58.9±3.7 | 75.3±1.1 | 85.8±2.5 | 19.8±2.0 | 19.2±1.5 | 5.0 (2.1) |
| GPR-GNN | 75.1±0.8 | 52.3±1.6 | 83.4±1.3 | 79.5±1.6 | 19.3±1.7 | 17.1±2.0 | 6.0 (1.1) |
| SLIMG | 85.7±0.8 | 67.8±2.8 | 87.9±1.0 | 85.1±1.3 | 82.5±1.6 | 31.1±1.1 | 3.3 (1.3) |
| NETINFOF | **87.3±0.7** | **86.7±0.6** | **89.8±0.3** | **89.8±1.0** | **89.6±0.2** | **90.8±0.7** | **1.0 (0.0)** |

## 6.2 NODE CLASSIFICATION

**Designs.** We remain the same scenarios in SLIMG, while using our graph generator in Appx. D.1.

**Observations.** Fig. 4b shows that NETINFOF_SCORE is highly correlated with test accuracy. In Appx. C.5, Table 10 shows that NETINFOF generalizes to all scenarios as SLIMG does; Table 11 shows that the component with the highest NETINFOF_SCORE always has the highest test accuracy.

## 7 EXPERIMENTS

We conduct experiments by real-world graphs to answer the following research questions (RQ):

RQ1. **Effectiveness:** How well does NETINFOF perform in real-world graphs?
RQ2. **Scalability:** Does NETINFOF scales linearly with the input size?
RQ3. **Ablation Study:** Are all the design choices in NETINFOF necessary?

The details of datasets and settings are in Appx. D. Since we focus on improving linear GNNs in link prediction, the experiments for node classification are in Appx. C.6 because of space limit. The experiments are conducted on an AWS EC2 G4dn instance with 192GB RAM.

### 7.1 EFFECTIVENESS (RQ1)

**Real-World Datasets.** We evaluate NETINFOF on 7 homophily and 5 heterophily real-world graphs. We randomly split the edges into training, validation, and testing sets by the ratio 70%/10%/20% five times and report the average for fair comparison. Since our goal is to propose a general GNN method, we focus on comparing NETINFOF with 6 GNN baselines, which are general GNNs (GCN, SAGE, GAT), heterophily GNNs (H²GCN, GPR-GNN), and a linear GNN (SLIMG). While Hits@100 is used for evaluating on most graphs, Hits@1000 is used on the larger ones, namely, Products, Twitch, and Pokec, which have much more negative edges in the testing sets.

In Table 3, NETINFOF outperforms GNN baselines in 11 out of 12 datasets, and has the highest average rank, as our derived embeddings include comprehensive graph information, that is, structure, features, and features propagated through structure. Compared to non-linear GNNs, SLIMG performs worse in most heterophily graphs, showing that it cannot properly measure the node similarity of heterophily embeddings in link prediction. By addressing the limitations of linear GNNs, NETINFOF is able to consistently outperform both SLIMG and non-linear GNNs in both homophily and heterophily graphs. Note that the results in Pokec are similar to the ones in homophily graphs, since it can be labeled as either homophily (by locality) or heterophily (by gender).

Table 3: **NETINFOF wins** on link prediction in most real-world datasets. Hits@100 is reported for most datasets, and Hits@1000 for the large datasets (Products, Twitch, and Pokec).

| Model | Cora | CiteSeer | PubMed | Comp. | Photo | ArXiv | Products | Cham. | Squirrel | Actor | Twitch | Pokec | Avg. Rank |
|---|---|---|---|---|---|---|---|---|---|---|---|---|---|
| GCN | 67.1±1.8 | 60.4±10. | 47.6±13. | 22.5±3.1 | 39.1±1.6 | 14.8±0.6 | 02.2±0.1 | 82.1±4.5 | 16.5±1.0 | 31.1±1.7 | 16.2±0.3 | 07.9±1.7 | 4.1 (1.3) |
| SAGE | 68.4±2.8 | 55.9±2.5 | 57.6±1.1 | 27.5±2.1 | 40.0±1.9 | 00.7±0.1 | 00.3±0.2 | 84.7±3.6 | 15.5±1.5 | 27.6±1.4 | 08.7±0.6 | 05.5±0.5 | 4.5 (1.2) |
| GAT | 66.7±3.6 | 65.2±2.6 | 55.1±2.4 | 28.3±1.6 | 44.2±3.5 | 05.0±0.8 | O.O.M. | 84.8±4.5 | 15.6±0.8 | 32.3±2.4 | 08.2±0.3 | O.O.M. | 4.0 (1.7) |
| H²GCN | 64.4±3.4 | 35.7±5.4 | 50.5±0.9 | 17.9±0.7 | 29.5±2.4 | O.O.M. | O.O.M. | 79.3±4.5 | 16.0±2.6 | 28.7±2.1 | O.O.M. | O.O.M. | 6.2 (1.0) |
| GPR-GNN | 69.8±1.9 | 53.5±8.1 | 66.3±3.3 | 20.7±1.8 | 34.1±1.1 | 13.8±0.8 | O.O.M. | 77.2±5.6 | 14.6±2.7 | 32.1±1.3 | 12.6±0.2 | 05.0±0.2 | 4.6 (1.8) |
| SLIMG | 77.9±1.3 | 86.8±1.0 | 55.9±2.8 | 25.3±0.9 | 40.2±2.5 | 20.2±1.0 | 27.6±0.6 | 76.9±2.8 | 19.6±1.5 | 18.7±1.0 | 12.0±0.3 | 21.7±0.2 | 3.5 (1.8) |
| NETINFOF | **81.3±0.6** | **87.3±1.3** | 59.7±1.1 | **31.1±1.9** | **46.8±2.2** | **39.2±1.8** | **35.2±1.1** | **86.9±2.3** | **24.2±2.0** | **36.2±1.2** | **19.6±0.7** | **31.3±0.5** | **1.1 (1.3)** |

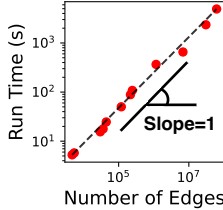

Figure 6: **NETINFOF_ACT is scalable** on link prediction, being linear with number of edges.

Table 4: **NETINFOF wins** on link prediction in OGB datasets. Hits@20, Hits@50, and Hits@100 are reported for ddi, collab, and ppa, respectively.

| Model | ogbl-ddi | ogbl-collab | ogbl-ppa |
|---|---|---|---|
| GCN | 37.1±5.1 | 44.8±1.1 | 18.7±1.3 |
| SAGE | 53.9±4.7 | 48.1±0.8 | 16.6±2.4 |
| SLIMG | 35.9±0.6 | 45.1±0.2 | 21.1±0.6 |
| NETINFOF | **56.8±3.4** | **53.7±0.2** | **24.2±0.1** |

**OGB Link Prediction Datasets.** We evaluate NETINFOF on OGB datasets. Table 4 shows that NETINFOF outperforms other general GNN baselines, while using a model with much fewer parameters. NETINFOF has only 1280 learnable parameters for all datasets, while GCN and SAGE have at least 279K and 424K, respectively, which is 218× more than the ones that NETINFOF has.

## 7.2 SCALABILITY (RQ2)

We plot the number of edges versus the run time of link prediction in seconds on the real-world datasets. In Fig. 6, we find that NETINFOF scales linearly with the number of edges, thanks to our speed-up techniques in estimating compatibility matrix $\mathbf{H}^*$. To give a concrete example, the numbers of coefficients of $\mathbf{H}_U^*$ are reduced from $d(d+1)/2 = 8256$ to 3208, 5373, and 4293, for Products, Twitch, and Pokec, respectively. Moreover, those numbers are very reasonable: Products is a homophily graph, its $\mathbf{H}_U^*$ has the fewest coefficients, which are mostly on the diagonal; Twitch is a heterophily graph, its $\mathbf{H}_U^*$ has the most coefficients, which are mostly on the off-diagonal; Pokec can be seen as either homophily or heterophily, its $\mathbf{H}_U^*$ has the number of coefficients in between.

## 7.3 ABLATION STUDY (RQ3)

To demonstrate the necessity of addressing the limitations of linear GNNs in link prediction with our design choices, we study NETINFOF (1) without compatibility matrix (w/o CM), and (2) with only compatibility matrix $\mathbf{H}$ (w/ only $\mathbf{H}$), which is not optimized with negative edges. Table 5 shows that NETINFOF works best with both design choices. In heterophily graphs, merely using $\mathbf{H}$ leads to better performance because of properly handling heterophily embeddings; while in homophily graphs, it accidentally increases the similarity between nodes in negative edges and hurts the performance. Taking into account both heterophily embeddings and negative edges, using $\mathbf{H}^*$ as the compatibility matrix has the best performance in both heterophily and homophily graphs.

Table 5: **Ablation Study - All design choices in NETINFOF are necessary** on link prediction. CM stands for compatibility matrix, and $\mathbf{H}$ is not optimized with negative edges.

| Model | Cora | CiteSeer | PubMed | Comp. | Photo | ArXiv | Products | Cham. | Squirrel | Actor | Twitch | Pokec |
|---|---|---|---|---|---|---|---|---|---|---|---|---|
| w/o CM | 79.8±0.9 | 86.5±1.6 | 58.5±1.2 | 27.9±0.2 | 44.7±3.2 | 35.9±1.8 | 34.6±0.4 | 74.6±1.5 | 14.3±0.6 | 29.5±1.8 | 08.9±0.8 | 30.5±0.3 |
| w/ only H | 80.9±0.5 | 87.0±1.4 | 58.8±1.4 | 26.5±1.2 | 43.4±2.1 | 32.5±1.6 | 30.6±0.4 | 74.3±3.2 | 19.3±1.2 | 32.2±1.6 | 10.3±3.1 | 29.8±0.4 |
| NETINFOF | **81.3±0.6** | **87.3±1.3** | **59.7±1.1** | **31.1±1.9** | **46.8±2.2** | **39.2±1.8** | **35.2±1.1** | **86.9±2.3** | **24.2±2.0** | **36.2±1.2** | **19.6±0.7** | **31.3±0.5** |

## 8 CONCLUSIONS

We propose the NETINFOF framework to measure and exploit the network usable information (NUI). In summary, NETINFOF has the following advantages:

1. *General*, handling both link prediction and node classification (Lemma 1-2);
2. *Principled*, with theoretical guarantee (Thm. 1-2) and closed-form solution (Lemma 1-2);
3. *Effective*, thanks to the proposed adjustment of node similarity;
4. *Scalable*, scaling linearly with the input size (Fig. 6).

Applied on our carefully designed synthetic datasets, NETINFOF correctly identifies the ground truth of NUI and is the only method that is robust to all graph scenarios. Applied on real-world graphs, NETINFOF wins in *11 out of 12* times on link prediction.

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

# A APPENDIX: PROOFS

## A.1 PROOF OF THM. 1

*Proof.* Let $Y$ be a discrete random variable with $n$ outcomes $(y_1, \ldots, y_n)$, and with probabilities $(p_1, \ldots, p_n)$, where:

$$1 = p_1 + \ldots + p_n \tag{9}$$

Let $p_{max}$ be the highest probability (break ties arbitrarily), that is:

$$p_{max} = \max_i p_i \tag{10}$$

For ease of presentation, and without loss of generality, assume that the most likely outcome is the first one, $y_1$, and thus $p_{max} = p_1$. Given no other information, the best classifier for $Y$ is the one that always guesses outcome $y_1$, and it has accuracy:

$$\text{accuracy}(Y) = p_1 \equiv p_{max} \tag{11}$$

The entropy $H(Y)$ is:

$$H(Y) = -(p_1 \log p_1 + \ldots + p_n \log p_n) \tag{12}$$

Thus we have:

$$2^{-H(Y)} = p_1{}^{p_1} * p_2{}^{p_2} \ldots * p_n{}^{p_n} \tag{13}$$

$$\leq p_{max}{}^{p_1} * p_{max}{}^{p_2} * \ldots * p_{max}{}^{p_n} \quad // \because p_{max} \geq p_i \tag{14}$$

$$\leq p_{max}{}^{p_1+p_2+\ldots+p_n} \tag{15}$$

$$\leq p_{max} \tag{16}$$

which completes the proof. ∎

## A.2 PROOF OF THM. 2

*Proof.* Let $Y$ and $X$ be two discrete random variables with $n$ outcomes $(y_1, \ldots, y_n)$ and $m$ outcomes $(x_1, \ldots, x_m)$, respectively, then their joint probabilities are $(p_{1,1}, \ldots, p_{m,n})$, where:

$$1 = p_1 + \ldots + p_m = \sum_{j=1}^{n} p_{1,j} + \ldots + \sum_{j=1}^{n} p_{m,j} \tag{17}$$

The accuracy is:

$$\text{accuracy}(Y|X) = \sum_{i=1}^{m} \max_j p_{i,j} \tag{18}$$

The conditional entropy $H(Y|X)$ is:

$$H(Y|X) = p_1 * \left(-\sum_{j=1}^{n} \frac{p_{1,j}}{p_1} * \log_2 \frac{p_{1,j}}{p_1}\right) + \ldots + p_m * \left(-\sum_{j=1}^{n} \frac{p_{m,j}}{p_m} * \log_2 \frac{p_{m,j}}{p_m}\right) \tag{19}$$

Thus we have:

$$\Rightarrow -H(Y|X) = p_1 * \left(\sum_{j=1}^{n} \frac{p_{1,j}}{p_1} * \log_2 \frac{p_{1,j}}{p_1}\right) + \ldots + p_m * \left(\sum_{j=1}^{n} \frac{p_{m,j}}{p_m} * \log_2 \frac{p_{m,j}}{p_m}\right) \tag{20}$$

$$\leq p_1 * \log_2 \left(\sum_{j=1}^{n} \left(\frac{p_{1,j}}{p_1}\right)^2\right) + \ldots + p_m * \log_2 \left(\sum_{j=1}^{n} \left(\frac{p_{m,j}}{p_m}\right)^2\right) \quad // \because \text{Jensen's Inequality} \tag{21}$$

$$\leq p_1 * \log_2 \left(p_1^{-2} * \sum_{j=1}^{n} p_{1,j}{}^2\right) + \ldots + p_m * \log_2 \left(p_m^{-2} * \sum_{j=1}^{n} p_{m,j}^2\right) \tag{22}$$

$$\leq p_1 * \log_2 \left(p_1^{-1} * \max_j p_{1,j}\right) + \ldots + p_m * \log_2 \left(p_m^{-1} * \max_j p_{m,j}\right) \tag{23}$$

This is because $\forall i = 1, \ldots, m$:

$$\sum_{j=1}^{n} p_{i,j}^2 = p_{i,1} * p_{i,1} + \ldots + p_{i,n} * p_{i,n} \tag{24}$$

$$\leq p_{i,1} * \max_j p_{i,j} + \ldots + p_{i,n} * \max_j p_{i,j} \tag{25}$$

$$\leq (p_{i,1} + \ldots + p_{i,n}) * \max_j p_{i,j} \tag{26}$$

$$= p_i * \max_j p_{i,j} \tag{27}$$

To continue, we have:

$$\Rightarrow 2^{-H(Y|X)} \leq 2^{p_1 * \log_2 (p_1^{-1} * \max_j p_{1,j}) + \ldots + p_m * \log_2 (p_m^{-1} * \max_j p_{m,j})} \tag{28}$$

$$\leq (p_1^{-1} * \max_j p_{1,j})^{p_1} * \ldots * (p_m^{-1} * \max_j p_{m,j})^{p_m} \tag{29}$$

$$\leq \max_j p_{1,j} + \ldots + \max_j p_{m,j} \quad // \because \text{Weighted AM-GM Inequality} \tag{30}$$

$$= \text{accuracy}(Y|X) \tag{31}$$

which completes the proof. ∎

### A.3 PROOF OF LEMMA 1

*Proof.* The goal is to maximize the similarity of nodes connected by edges. If we start from cosine similarity and L2-normalize the node embeddings $\mathbf{z}$, we have:

$$\begin{aligned}
&\frac{\mathbf{z}_i \boldsymbol{H} \mathbf{z}_j^\mathsf{T}}{\|\mathbf{z}_i\| \|\mathbf{z}_j\|} = 1, \forall (i,j) \in \mathcal{E} \\
&\Rightarrow \mathbf{z}_i \boldsymbol{H} \mathbf{z}_j^\mathsf{T} = \|\mathbf{z}_i\| \|\mathbf{z}_j\| = 1 \\
&\Rightarrow \mathbf{z}_i \boldsymbol{H} = \mathbf{z}_j
\end{aligned} \tag{32}$$

Setting $\mathbf{z}_i$ as the input data and $\mathbf{z}_j$ as the target data, this equation can be solved by $d$-target linear regression with $d$ coefficients, which has the closed-form solution. ∎

### A.4 PROOF OF LEMMA 2

*Proof.* First, we rewrite the adjusted node similarity $s$ from matrix form into a simple computation:

$$\begin{aligned}
s(\mathbf{z}_i, \mathbf{z}_j, \boldsymbol{H}) &= \mathbf{z}_i \boldsymbol{H} \mathbf{z}_j^\mathsf{T} \\
&= [\mathbf{z}_{i,1} \quad \cdots \quad \mathbf{z}_{i,d}] \begin{bmatrix} \boldsymbol{H}_{1,1} & \cdots & \boldsymbol{H}_{1,d} \\ \vdots & \ddots & \vdots \\ \boldsymbol{H}_{d,1} & \cdots & \boldsymbol{H}_{d,d} \end{bmatrix} \begin{bmatrix} \mathbf{z}_{j,1} \\ \vdots \\ \mathbf{z}_{j,d} \end{bmatrix} \\
&= \begin{bmatrix} \mathbf{z}_{i,1}\mathbf{z}_{j,1} & \cdots & \mathbf{z}_{i,1}\mathbf{z}_{j,d} \\ \vdots & \ddots & \vdots \\ \mathbf{z}_{i,d}\mathbf{z}_{j,1} & \cdots & \mathbf{z}_{i,d}\mathbf{z}_{j,d} \end{bmatrix} \odot \begin{bmatrix} \boldsymbol{H}_{1,1} & \cdots & \boldsymbol{H}_{1,d} \\ \vdots & \ddots & \vdots \\ \boldsymbol{H}_{d,1} & \cdots & \boldsymbol{H}_{d,d} \end{bmatrix} \\
&= \mathbf{z}_{i,1}\mathbf{z}_{j,1}\boldsymbol{H}_{1,1} + \mathbf{z}_{i,1}\mathbf{z}_{j,2}\boldsymbol{H}_{1,2} + \cdots + \mathbf{z}_{i,1}\mathbf{z}_{j,d}\boldsymbol{H}_{1,d} + \cdots + \mathbf{z}_{i,d}\mathbf{z}_{j,d}\boldsymbol{H}_{d,d}
\end{aligned} \tag{33}$$

Next, to maximize the similarity of nodes connected by positive edges, and to minimize the similarity of nodes connected by negative edges, we have:

$$s(\mathbf{z}_i, \mathbf{z}_j, \boldsymbol{H}) = \begin{cases} 1 & (i,j) \in \mathcal{E} \\ 0 & (i,j) \in \mathcal{E}_{\text{neg}} \end{cases} \tag{34}$$

Therefore, this equation can be solved by linear regression with $d^2$ coefficients, which has the closed-form solution.

∎

### A.5 PROOF OF LEMMA 3

*Proof.* NETINFOF includes four parts: SVD, PCA, compatibility matrix estimation, and LogitReg. The time complexity of truncated SVD is $O(d^2|\mathcal{V}|)$, and the one of PCA is $f^2|\mathcal{V}|+f^3$. Compatibility matrix estimation is optimized by Ridge regression with regularized least-squares routine, whose time complexity is $d^4|\mathcal{E}|$. The time complexity of training LogitReg is $dt|\mathcal{E}|$, where $t$ is the number of epochs. In our experiments, $t$ is no greater than 100, and $|\mathcal{V}|$ is much less than $|\mathcal{E}|$ in most datasets. By combining all the terms and keeping only the dominant ones, we have time complexity of NETINFOF for link prediction $O(f^2|\mathcal{V}| + f^3 + d^4|\mathcal{E}|)$. ∎

## B APPENDIX: ALGORITHMS

---

**Algorithm 1:** Compatibility Matrix with Negative Edges

---

**Input:** Preprocessed node embedding $\hat{\mathbf{Z}}$, edge set $\mathcal{E}$, and sample size $S$

1 Extract 2-core edge set $\mathcal{E}_{pos}$ from $\mathcal{E}$;
2 Estimate $\mathbf{H}$ with $\hat{\mathbf{Z}}$ and $\mathcal{E}_{pos}$ by Lemma 1;
3 **if** $|\mathcal{E}_{pos}| > S$ **then**
4 $\quad$ Sample $S$ edges from $\mathcal{E}_{pos}$;
5 Initialize $\mathbf{H}^*$ with $\mathbf{H}$;
6 Keep top coefficients in upper triangle of $\mathbf{H}^*$ with 95% energy;
7 Sample $2|\mathcal{E}_{pos}|$ negative edges as $\mathcal{E}_{neg}$;
8 Estimate $\mathbf{H}^*$ with $\hat{\mathbf{Z}}$, $\mathcal{E}_{pos}$ and $\mathcal{E}_{neg}$ by Lemma 2;
9 Return $\mathbf{H}^*$;

---

**Algorithm 2:** NETINFOF_PROBE for Link Prediction

---

**Input:** Node embedding $\mathbf{Z}$, train edge set $\mathcal{E}_{train}$, valid edge set $\mathcal{E}_{valid}$, valid edge labels $\mathbf{y}_{valid}$, sample size $S$, and bin size $k$

1 Preprocess $\mathbf{Z}$ into $\hat{\mathbf{Z}}$ by column-wise standardization and row-wise L2-normalization;
2 $\mathbf{H}^* = $ Compatibility-Matrix-with-Negative-Edges($\hat{\mathbf{Z}}, \mathcal{E}_{train}, S$);
3 **if** $|\mathcal{E}_{pos}| > S$ **then**
4 $\quad$ Sample $S$ edges from $\mathcal{E}_{pos}$;
5 **else**
6 $\quad$ $\mathcal{E}_{pos} \leftarrow \mathcal{E}$;
7 Sample $2|\mathcal{E}_{pos}|$ negative edges as $\mathcal{E}_{neg}$;
8 Fit $k$-bins discretizer with $\hat{\mathbf{z}}_i\mathbf{H}^*\hat{\mathbf{z}}_j^\mathsf{T}, \forall(i,j) \in \mathcal{E}_{pos} \cup \mathcal{E}_{neg}$;
9 Discretize $\hat{\mathbf{z}}_i\mathbf{H}^*\hat{\mathbf{z}}_j^\mathsf{T}, \forall(i,j) \in \mathcal{E}_{valid}$ into $k$ bins as $\mathbf{s}_{valid}$;
10 Return NETINFOF_SCORE, computed with $\mathbf{s}_{valid}$ and $\mathbf{y}_{valid}$ by Eq. 2;

---

**Algorithm 3:** NETINFOF_PROBE for Node Classification

---

**Input:** Train, valid, and test node embedding $\mathbf{Z}_{train}$, $\mathbf{Z}_{valid}$, and $\mathbf{Z}_{test}$, train and valid node labels $\mathbf{y}_{train}$ and $\mathbf{y}_{valid}$, and cluster number $k$

1 Preprocess $\mathbf{Z}$ into $\hat{\mathbf{Z}}$ row-wise L2-normalization;
2 Fit clustering model with $\hat{\mathbf{Z}}_{test}$;
3 Assign cluster labels $\mathbf{s}_{train}$ and $\mathbf{s}_{valid}$ to $\hat{\mathbf{Z}}_{train}$ and $\hat{\mathbf{Z}}_{valid}$, respectively;
4 Return NETINFOF_SCORE, computed with $\mathbf{s}_{train} \cup \mathbf{s}_{valid}$ and $\mathbf{y}_{train} \cup \mathbf{y}_{valid}$ by Eq. 2;

---

## C APPENDIX: EXPERIMENTS

### C.1 LINK PREDICTION IN SYNTHETIC DATASETS

To study the information of the derived node embeddings, we conduct NETINFOF_PROBE on each component, and LogitReg to have test performance on link prediction. In Table 6, the components with the top-2 NETINFOF_SCORE as well have the top-2 test performance in every scenario.

Table 6: Results of each derived node embeddings. NETINFOF_SCORE and test Hits@100 on link prediction are reported. Red highlights the results close to random guessing.

| Metric | Feature Component | Random X Diagonal A | Random X Off-Diag. A | Global X Diagonal A | Global X Off-Diag. A | Local X Diagonal A | Local X Off-Diag. A |
|---|---|---|---|---|---|---|---|
| NETINFOF_SCORE | C1 : U | 75.1±0.9 | 74.3±0.5 | 75.3±0.6 | 74.1±0.7 | 75.2±0.8 | 74.8±0.5 |
| | C2 : R | **76.5±0.9** | **76.3±0.5** | 76.7±0.8 | 76.3±0.2 | 76.6±0.8 | 76.3±0.4 |
| | C3 : F | 50.0±0.0 | 50.0±0.0 | 67.0±0.5 | 71.5±0.6 | 75.9±1.1 | 75.5±0.6 |
| | C4 : P | 75.1±0.7 | 73.2±0.6 | 78.2±0.7 | 78.5±0.5 | 78.5±1.2 | 78.9±0.5 |
| | C5 : S | 74.3±1.1 | 72.0±0.5 | **78.7±1.0** | **79.2±0.5** | **79.0±0.9** | **81.1±0.6** |
| Test Hits@100 | C1 : U | **83.3±1.2** | **76.4±2.1** | 83.5±1.2 | 76.7±0.5 | 83.2±1.1 | 78.2±0.9 |
| | C2 : R | 83.0±1.5 | 74.7±1.4 | 83.0±1.3 | 75.0±1.5 | 83.2±1.0 | 75.3±0.7 |
| | C3 : F | 02.2±0.2 | 02.5±0.2 | 52.3±1.9 | 63.0±2.2 | 80.7±1.1 | 77.1±1.0 |
| | C4 : P | 82.3±1.2 | 73.4±0.4 | 86.2±1.1 | **79.3±0.7** | 85.7±1.5 | 79.4±1.3 |
| | C5 : S | 80.7±0.9 | 68.1±1.8 | **86.4±0.9** | 79.2±1.1 | **86.5±1.0** | **85.2±1.1** |

### C.2 LINK PREDICTION IN REAL-WORLD DATASETS

To study the effectiveness of each derived node embedding, we conduct experiments on each individual component for the real-world datasets. As shown in Table 7, different components have variable impact depending on the input graph.

Table 7: **Ablation study** - NETINFOF on each component of derived node embedding. Hits@100 is reported for most datasets, and Hits@1000 for the large datasets (Products, Twitch, and Pokec).

| Model | Cora | CiteSeer | PubMed | Comp. | Photo | ArXiv | Products | Cham. | Squirrel | Actor | Twitch | Pokec |
|---|---|---|---|---|---|---|---|---|---|---|---|---|
| C1 : U | 48.3±0.8 | 36.9±2.5 | 43.7±1.8 | 24.0±1.8 | 38.5±2.6 | 18.1±0.6 | 13.2±0.3 | 75.0±5.2 | 12.3±1.8 | 23.2±2.2 | **15.8±0.2** | **16.2±0.6** |
| C2 : R | 61.5±1.2 | 50.0±1.9 | 47.9±0.8 | 19.4±0.8 | 36.8±3.3 | 12.3±1.0 | 09.4±0.7 | 64.6±3.2 | 08.2±1.4 | **34.2±2.3** | 01.5±0.2 | 05.2±0.2 |
| C3 : F | 58.3±3.2 | 71.5±2.4 | 41.6±0.4 | 07.1±0.4 | 15.6±1.2 | 04.9±0.2 | 00.4±0.1 | 10.7±1.0 | 00.6±0.1 | 02.6±0.5 | 01.7±0.6 | 00.5±0.2 |
| C4 : P | 67.3±1.8 | 60.9±2.5 | 48.1±1.8 | **28.6±2.4** | **42.4±1.4** | **34.2±1.0** | 27.6±0.5 | **84.3±1.8** | **20.9±2.3** | 13.1±1.1 | 07.3±1.0 | 12.4±0.8 |
| C5 : S | **82.4±1.0** | **88.7±1.5** | **63.3±1.1** | 29.0±2.3 | 40.3±1.1 | 33.6±1.1 | **28.1±0.7** | 80.5±2.7 | 19.2±1.2 | 22.3±0.9 | 02.7±1.8 | **16.2±0.6** |

### C.3 SENSITIVITY ANALYSIS OF NETINFOF_PROBE IN LINK PREDICTION

We conduct a sensitivity analysis on a medium size dataset "Computers", to study the effect of the number of bins $k$ in NETINFOF_PROBE (Algo. 2). As shown in Table 8 and Fig. 7, NETINFOF_SCORE is insensitive to the exact value of $k$, forming a plateau when $k$ increases.

Table 8: **Sensitivity analysis** of NETINFOF_PROBE in link prediction. NETINFOF_SCORE is reported.

| Number of Bins | k = 4 | k = 8 | k = 16 | k = 32 | k = 64 |
|---|---|---|---|---|---|
| C1 : U | 74.2±0.3 | 79.7±0.2 | 80.4±0.3 | 80.8±0.3 | 81.0±0.3 |
| C2 : R | 76.5±0.2 | 80.4±0.0 | 81.2±0.3 | 81.4±0.3 | 81.5±0.3 |
| C3 : F | 63.8±0.1 | 64.9±0.2 | 65.0±0.1 | 65.0±0.1 | 65.1±0.1 |
| C4 : P | 77.3±0.2 | 82.1±0.1 | 83.0±0.3 | 83.3±0.3 | 83.4±0.3 |
| C5 : S | 77.3±0.1 | 82.5±0.1 | 83.3±0.1 | 83.6±0.2 | 83.7±0.2 |

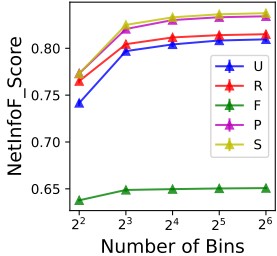

Figure 7: **Sensitivity analysis** - NETINFOF_PROBE is insensitive to the number of bins $k$.

## C.4 COMPARISON WITH SUBGRAPH GNN IN REAL-WORLD DATASETS

Although the comparison with subgraph GNN is beyond the scope of this paper, the experimental results of a subgraph GNN, SEAL (Zhang & Chen, 2018), are provided in Table 9 for the interested readers. The hidden size is set to $128$ as the same as other baselines, and the hyperparameter search is done on the number of hops ($[1, 3]$), learning rate ($[0.001, 0.01]$), weight decay ($[0, 5e^{-4}]$), and the number of layers ($[2, 3]$). Other hyperparameters are set as the default ones in Zhang et al. (2021).

To emphasize again, as shown in Table 1, subgraph GNNs are beyond the scope of this paper, because: (1) they are not designed as general GNNs and can only solve link prediction; (2) they are only scalable to small graphs. To give a concrete example, using an AWS EC2 G4dn instance with $384$GB RAM and running on the largest dataset Photo without running out of memory (O.O.M.) in the preprocessing step, SEAL takes $17606$ seconds (around $4.9$ hours) to train on a single split of data, while NETINFOF takes only $52$ seconds, which is $\mathbf{339\times}$ faster than SEAL.

Table 9: Comparison with SEAL. Hits@100 is reported for most datasets, and Hits@1000 for the large datasets (Products, Twitch, and Pokec).

| Model | Cora | CiteSeer | PubMed | Comp. | Photo | ArXiv | Products | Cham. | Squirrel | Actor | Twitch | Pokec |
|---|---|---|---|---|---|---|---|---|---|---|---|---|
| SEAL | 66.4±0.6 | 61.4±1.0 | **62.8±1.7** | O.O.M. | **55.5±1.0** | O.O.M. | O.O.M. | **89.4±1.0** | O.O.M. | **43.6±0.7** | O.O.M. | O.O.M. |
| NETINFOF | **81.3±0.6** | **87.3±1.3** | 59.7±1.1 | **31.1±1.9** | 46.8±2.2 | **39.2±1.8** | **35.2±1.1** | 86.9±2.3 | **24.2±2.0** | 36.2±1.2 | **19.6±0.7** | **31.3±0.5** |

## C.5 NODE CLASSIFICATION IN SYNTHETIC DATASETS

We keep the same settings of sanity checks in SLIMG, but use our graph generation process in Appx. D.1. To study the information of the derived node embeddings, we conduct NET-INFOF_PROBE on each component, and LogitReg to have test performance on node classification. In Table 11, the component with the highest NETINFOF_SCORE as well has the highest test accuracy in every scenario. In Table 10, NETINFOF generalizes to all scenarios as SLIMG does.

Table 10: **NETINFOF works well** on node classification in synthetic datasets. Accuracy is reported.

| Model | Useful X Uniform A | Random X Homophily A | Random X Heterophily A | Useful X Homophily A | Useful X Heterophily A |
|---|---|---|---|---|---|
| SLIMG | 85.4±2.3 | 88.9±0.4 | 87.4±2.5 | 97.3±0.2 | 97.0±0.1 |
| NETINFOF | 86.7±1.5 | 88.6±1.5 | 87.9±1.5 | 97.1±0.1 | 97.0±0.2 |

Table 11: Results of each derived node embeddings. NETINFOF_SCORE, and test accuracy on node classification are reported. Red highlights the results close to random guessing.

| Metric | Feature Component | Useful X Uniform A | Random X Homophily A | Random X Heterophily A | Useful X Homophily A | Useful X Heterophily A |
|---|---|---|---|---|---|---|
| NETINFOF_SCORE | $C1 : U$ | 25.9±0.2 | **73.8±2.8** | **74.4±3.9** | 73.6±2.6 | 75.4±4.5 |
| | $C2 : R$ | 25.7±0.1 | 62.5±3.5 | 56.7±1.3 | 64.7±3.7 | 56.3±3.3 |
| | $C3 : F$ | **64.6±4.2** | 26.0±0.4 | 26.0±0.4 | 73.1±3.2 | 73.1±3.2 |
| | $C4 : P$ | 33.7±1.5 | 56.0±4.2 | 58.9±3.9 | **87.2±3.6** | 87.2±3.2 |
| | $C5 : S$ | 33.9±0.8 | 58.7±4.8 | 55.5±3.5 | 86.1±2.4 | **88.3±3.4** |
| Test Accuracy | $C1 : U$ | 25.1±0.3 | **80.3±0.9** | **81.3±0.8** | 80.3±0.9 | 81.3±0.8 |
| | $C2 : R$ | 25.3±0.4 | 60.7±0.8 | 58.4±1.4 | 61.5±0.9 | 58.8±1.3 |
| | $C3 : F$ | **74.7±0.8** | 25.4±0.6 | 25.4±0.6 | 74.7±1.1 | 74.7±1.1 |
| | $C4 : P$ | 47.9±0.9 | 75.2±0.9 | 77.4±0.8 | **97.1±0.1** | 96.6±0.1 |
| | $C5 : S$ | 47.9±0.9 | 75.8±1.0 | 74.1±0.8 | 96.9±0.1 | **96.9±0.1** |

## C.6 NODE CLASSIFICATION IN REAL-WORLD DATASETS

We follow the same experimental settings in SLIMG. The nodes are randomly split by the ratio $2.5\%/2.5\%/95\%$ into the training, validation and testing sets. In Table 12, we find that NETINFOF always wins and ties with the top baselines.

Table 12: **NETINFOF works well** on node classification in real-world datasets. Accuracy is reported.

| Model | Cora | CiteSeer | PubMed | Comp. | Photo | ArXiv | Products | Cham. | Squirrel | Actor | Twitch | Pokec |
|---|---|---|---|---|---|---|---|---|---|---|---|---|
| GCN | 76.0±1.2 | 65.0±2.9 | 84.3±0.5 | 85.1±0.9 | 91.6±0.5 | 62.8±0.6 | O.O.M. | 38.5±3.0 | **31.4±1.8** | 26.8±0.4 | 57.0±0.1 | 63.9±0.4 |
| SAGE | 74.6±1.3 | 63.7±3.6 | 82.9±0.4 | 83.8±0.5 | 90.6±0.5 | 61.5±0.6 | O.O.M. | 39.8±4.3 | 27.0±1.3 | 27.8±0.9 | 56.6±0.4 | 68.9±0.1 |
| H²GCN | 77.6±0.9 | 64.7±3.8 | **85.4±0.4** | 49.5±16. | 75.8±11. | O.O.M. | O.O.M. | 31.9±2.6 | 25.0±0.5 | 28.9±0.6 | 58.7±0.0 | O.O.M. |
| GPR-GNN | **78.8±1.3** | 64.2±4.0 | 85.1±0.7 | 85.0±1.0 | **92.6±0.3** | 58.5±0.8 | O.O.M. | 31.7±4.7 | 26.2±1.6 | 29.5±1.1 | 57.6±0.2 | 67.6±0.1 |
| GAT | 78.2±1.2 | 65.8±4.0 | 83.6±0.2 | 85.4±1.4 | 91.7±0.5 | 58.2±1.0 | O.O.M. | 39.1±4.1 | 28.6±0.6 | 26.4±0.4 | O.O.M. | O.O.M. |
| SLIMG | 77.8±1.1 | **67.1±2.3** | 84.6±0.5 | 86.3±0.7 | 91.8±0.5 | 66.3±0.3 | 84.9±0.0 | 40.8±3.2 | 31.1±0.7 | **30.9±0.6** | 59.7±0.1 | 73.9±0.1 |
| NETINFOF | 77.5±1.3 | 64.5±3.6 | 84.1±0.5 | **86.6±0.6** | 91.6±0.3 | **66.8±0.7** | **85.3±0.0** | **41.6±3.2** | 30.4±1.5 | 30.7±0.3 | **61.0±0.2** | **74.0±0.1** |

# D APPENDIX: REPRODUCIBILITY

## D.1 SYNTHETIC DATASETS

The synthetic datasets are composed of two parts, namely, graph structure and node features. Noting that link prediction and node classification share the same generator of graph structure, but differ in the one of node features. Noises are randomly injected into both graph structure and node features to ensure that they contain consistent usable information across different scenarios. The number of nodes is set to be 4000, and the number of features is set to be 800.

**Graph Structure.** There are three kinds of graph structures, namely, diagonal, off-diagonal, and uniform. For each graph, we equally assign labels with $c$ classes to all nodes. In the diagonal structure, the nodes are connected to the nodes with the same class label, which exhibits homophily. In the off-diagonal structure, the nodes are connected to the nodes with one different class label, which exhibits heterophily. In the uniform structure, the connections are randomly made between nodes. Other than randomly picking node pairs to make connections, we mimic the phenomenon that the nodes are connected by higher-order structure in the real-world (Eswaran et al., 2020). To achieve that, in the diagonal structure, a random amount of nodes (between 4 and 8) with the same class are randomly picked and made into a clique; In the off-diagonal structure, a random amount of nodes (between 4 and 8) from each of the two classes are randomly picked and made into a bipartite clique. This process continues until the graph reaches our desired density. In the link prediction, the assigned node labels are not used; in the node classification, they are used as the target labels.

**Node Features in Link Prediction.** In the case that the node features are useful, they are generated by the left singular vectors of the 2-step random walk matrix. The $(i, j)$ element of the matrix is the counting of the node on row $i$ visited the node on column $j$, and each node walks for 1000 trials. The node features are directly used in the global scenarios, but split into different slices based on the labels in the local scenarios. The random node features are the rows in a random binary matrix.

**Node Features in Node Classification.** In the case that the node features are useful, we randomly sample a center for each class label. For nodes with the same class, we add Gaussian noises on top of their class center. The random node features are the rows in a random binary matrix.

## D.2 REAL-WORLD DATASETS

In the experiments, we use 7 homophily and 5 heterophily real-world datasets that have been widely used before. All the graphs are made undirected and their statistics are reported in Table 13. We also conduct experiments on 3 link prediction datasets from Open Graph Benchmark (OGB) (Hu et al., 2020), namely ogbl-ddi[1], ogbl-collab[2], and ogbl-ppa[3].

**Homophily Graphs.** Cora (Motl & Schulte, 2015)[4], CiteSeer (Rossi & Ahmed, 2015)[5], and PubMed (Courtesy of the US National Library of Medicine, 1996)[6] are citation networks between

---

[1]https://ogb.stanford.edu/docs/linkprop/#ogbl-ddi

[2]https://ogb.stanford.edu/docs/linkprop/#ogbl-collab

[3]https://ogb.stanford.edu/docs/linkprop/#ogbl-ppa

[4]https://relational.fit.cvut.cz/dataset/CORA

[5]https://linqs.org/datasets/#citeseer-doc-classification

[6]https://www.nlm.nih.gov/databases/download/pubmed_medline.html

research articles. Computers and Photo (Ni et al., 2019)[7] are Amazon co-purchasing networks between products. ogbn-arXiv and ogbn-Products are large graphs from OGB (Hu et al., 2020). ogbn-arXiv[8] is a citation network of papers from arXiv; and ogbn-Products[9] is also an Amazon product co-purchasing network. In the node classification task, we omit the classes with instances fewer than 100 so that each class has enough training data.

**Heterophily Graphs.** Chameleon and Squirrel (Rozemberczki et al., 2021)[10] are Wikipedia page-to-page networks between articles from Wikipedia. Actor (Pei et al., 2020)[11] is a co-occurrence network of actors on Wikipedia pages. Twitch (Rozemberczki et al., 2021)[12] and Pokec (Takac & Zabovsky, 2012; Leskovec & Krevl, 2014)[13] are online social networks, which are relabeled by Lim et al. (2021)[14] to present heterophily. Penn94 is not included in this paper because of legal issues.

Table 13: Network Statistics. The left and right parts are homophily and heterophily, respectively.

| | Cora | CiteSeer | PubMed | Computers | Photo | ogbn-arXiv | ogbn-Products | Chameleon | Squirrel | Actor | Twitch | Pokec |
|---|---|---|---|---|---|---|---|---|---|---|---|---|
| # of Nodes | 2,708 | 3,327 | 19,717 | 13,752 | 7,650 | 169,343 | 2,449,029 | 2,277 | 5,201 | 7,600 | 168,114 | 1,632,803 |
| # of Edges | 5,429 | 4,732 | 44,338 | 245,861 | 119,081 | 1,166,243 | 61,859,140 | 36,101 | 216,933 | 29,926 | 6,797,557 | 30,622,564 |
| # of Features | 1433 | 3703 | 500 | 767 | 745 | 128 | 100 | 2325 | 2089 | 931 | 7 | 65 |
| # of Classes | 7 | 6 | 3 | 10 | 8 | 40 | 39 | 5 | 5 | 5 | 2 | 2 |

## D.3 EXPERIMENTAL SETTINGS

For fair comparison, each experiment is run with 5 different splits of both synthetic and real-world datasets. In link prediction, edges are split into training, validation and testing sets with the $70\%/10\%/20\%$ ratio. In node classification, the nodes are split into training, validation and testing sets with the $2.5\%/2.5\%/95\%$ ratio. For small graphs, the linear models are trained by L-BFGS for 100 epochs with the patience of 5, and the non-linear models are trained by ADAM for 1000 epochs with the patience of 200. For large graphs (Products, Twitch, and Pokec), most models are trained by ADAM for 100 epochs with the patience of 10, except GPR-GNN and GAT, they are trained by ADAM for 20 epochs with the patience of 5 to speedup. All the training are full-batch, and the same amount of negative edges are randomly sampled for each batch while training.

## D.4 HYPERPARAMETERS

Table 14: Search space of hyperparameters.

| Method | Hyperparameters |
|---|---|
| GCN | $lr = [0.001, 0.01], wd = [0, 5e^{-4}], layers = 2$ |
| SAGE | $lr = [0.001, 0.01], wd = [0, 5e^{-4}], layers = 2$ |
| H$_2$GCN | $lr = [0.001, 0.01], wd = [0, 5e^{-4}], layers = [1, 2]$ |
| GPR-GNN | $lr = [0.001, 0.01], wd = [0, 5e^{-4}], layers = 10, \alpha = [0.1, 0.2, 0.5, 0.9]$ |
| GAT | $lr = [0.001, 0.01], wd = [0, 5e^{-4}], layers = 2, heads = [8, 16]$ |
| SLIMG | $lr = 0.1, wd_1 = [1e^{-4}, 1e^{-5}], wd_2 = [1e^{-3}, 1e^{-4}, 1e^{-5}, 1e^{-6}]$ |
| NETINFOF | $lr = 0.1, wd_1 = [1e^{-4}, 1e^{-5}], wd_2 = [1e^{-3}, 1e^{-4}, 1e^{-5}, 1e^{-6}]$ |

The search space of the hyperparameters is provided in Table 14. Each experiment is run with 5 different splits of the dataset, and grid search of hyperparameters based on the validation performance is done on each of the splits. The hidden size of all methods is set to 128.

**Linear GNNs.** For sparse-group LASSO, $wd_1$ is the coefficient of overall sparsity, and $wd_2$ is the one of group sparsity. The derived embedding **R** in NETINFOF uses $T = 1000$ for the synthetic and OGB datasets, and $T = 200$ for the real-world datasets. The sample size $S$ is set to be $200,000$ in all experiments. Since the large graphs (Products, Twitch, and Pokec) have no more than 128 features, to ensure NETINFOF and SLIMG have enough parameters, we concatenate the one-hot node degree to the original features. The hidden size is set to 256 for NETINFOF and SLIMG in the OGB experiments. For ogbl-ddi, since there is no node features, they use the one-hot node degree as the node features. For ogbl-ppa, we concatenate the embedding from node2vec (Grover & Leskovec, 2016) to the original features, as Hamilton et al. (2017) did for GCN and SAGE.

---

[7] https://nijianmo.github.io/amazon/index.html

[8] https://ogb.stanford.edu/docs/nodeprop/#ogbn-arxiv

[9] https://ogb.stanford.edu/docs/nodeprop/#ogbn-products

[10] https://github.com/benedekrozemberczki/MUSAE/

[11] https://github.com/graphdml-uiuc-jlu/geom-gcn/tree/master/new_data/film

[12] https://github.com/benedekrozemberczki/datasets#twitch-social-networks

[13] https://snap.stanford.edu/data/soc-Pokec.html

[14] https://github.com/CUAI/Non-Homophily-Large-Scale

