# OpenReview forum: "NetInfoF Framework: Measuring and Exploiting Network Usable Information"
_ICLR.cc/2024/Conference — ICLR 2024 spotlight_

### Official Review · Reviewer_ANgE · 2023-10-26

**Soundness:** 2 fair
**Presentation:** 2 fair
**Contribution:** 2 fair
**Rating:** 6
**Confidence:** 4

**Summary:**

The paper discusses the NETINFOF framework, which focuses on measuring and utilizing the usable information in node-attributed graphs for graph tasks such as link prediction and node classification. The authors propose two components of the framework: NETINFOF PROBE, which measures the amount of information present in the graph structure and node features without any model training, and NETINFOF ACT, which uses the measured usable information to solve the graph tasks.

**Strengths:**

Lots of experiments

**Weaknesses:**

See below

**Questions:**

1. "How to connect the graph information with the performance metric on the task?" Here are some missed related works [1,2,3,4,5].

2. What is $f$ in C4? I suggest not using $f$, because you already use it as the number of features.

3. In theorem 1, "Given a discrete random variable Y , we have..." What is "accuracy(Y)"? Accuracy of a a discrete random variable? This looks weird and I suggest giving a more strict and accurate statement. What is $p_y$?

4. "Predicting links by GNNs relies on measuring node similarity, which is incorrect if the neighbors have heterophily embeddings." The embedding will be similar even with heterophily connections if the nodes share similar neighborhood patterns [6].

5. "the node embeddings of linear GNNs require no model training“ I don't understand this. Why linear GNNs don't require training? Do you mean SGC [7] don't require training?

6. "It results in low value when node i and node j have heterophily embeddings, even if they are connected by an edge." This is not correct, see [2,3,6].



[1] Characterizing graph datasets for node classification: Beyond homophily-heterophily dichotomy. arXiv preprint arXiv:2209.06177.

[2] Revisiting heterophily for graph neural networks. Advances in neural information processing systems, 35, 1362-1375.

[3] When do graph neural networks help with node classification: Investigating the homophily principle on node distinguishability. arXiv preprint arXiv:2304.14274.

[4] Demystifying Structural Disparity in Graph Neural Networks: Can One Size Fit All?. arXiv preprint arXiv:2306.01323.

[5] Exploiting Neighbor Effect: Conv-Agnostic GNN Framework for Graphs With Heterophily. IEEE Transactions on Neural Networks and Learning Systems.

[6] Is Homophily a Necessity for Graph Neural Networks?. In International Conference on Learning Representations 2022.

[7] Simplifying graph convolutional networks. In International conference on machine learning, pp. 6861–6871. PMLR, 2019.

---

> ### Author Response · Authors · 2023-11-16
> **Response to Reviewer ANgE**
>
> We thank the reviewer for the comments, and apologize for a few unclear writings that make the reviewer misunderstand the idea that we tried to express.
> We sincerely hope that the reviewer can re-evaluate our paper and raise the rating after reviewing our answers and revised paper.
> We address the questions one by one.
>
> > W1. "How to connect the graph information with the performance metric on the task?" Here are some missed related works [1,2,3,4,5].
>
> Thank you for your supplement on related works.
> We have added them to Section 2 (Page 2) in our revised paper.
> Noting that our focus has two major differences from [1, 2, 3, 4, 5]:
> 1. These works focus on node classification, while our work majorly focuses on **link prediction**.
> 2. These works focus on graphs that are heterophily defined by the node labels, while our work focuses on cases such as heterophily w.r.t. node features ("talkative people have silent friends"), as well as heterophily w.r.t. graph structure (forming bipartite cores). We did not consider node labels in link prediction.
>
> > W2. What is $f$ in C4?
>
> As mentioned in the last sentence in C4 (Section 3.2, Page 3), $f$ is the column-wise L2 normalization.
> We replace it with $l$ in our revised paper for increased clarity.
>
> > W3. In theorem 1, "Given a discrete random variable Y , we have..." What is "accuracy(Y)"?
>
> As shown in Equation 1 (Page 4), given a discrete random variable, its accuracy is the probability of its most probable outcome.
> This is achieved by majority voting, which is the best we can do without any other information.
> It is also described in Appendix A.1 (Page 13).
>
> > W4. "Predicting links by GNNs relies on measuring node similarity, which is incorrect if the neighbors have heterophily embeddings." The embedding will be similar even with heterophily connections if the nodes share similar neighborhood patterns.
>
> The main point is that we use **structure** embeddings (C1, Section 3.2, Page 3).
>
> We have clarified our wording as follows (Page 5, 4 lines from the top):
> *“Predicting links by GNNs relies on measuring node similarity, which is incorrect if the neighbors are expected to have dissimilar embeddings; for example, in a bipartite graph, …”* in our revised paper.
>
> We agree with the reviewer that in the (label-)heterophily graphs, the **propagated** embeddings (C4 and C5) of connected nodes will be proximity embeddings, and are very likely to be similar.
> However, for a (structure-)heterophily graph (e.g., consisting of bipartite cores), our proposed **structure** embeddings (C1), will not be similar even if they share similar neighborhood patterns.
>
> > W5. "the node embeddings of linear GNNs require no model training“. Do you mean SGC [7] don't require training?
>
> The node embeddings of linear GNNs in Equations 6 and 8 (Page 6 and 7, respectively) require no training, as they do not have any learnable parameters and can be computed with the closed-form formula.
> By “linear GNNs”, we mean GNNs that have linear transfer functions (instead of non-linear ones, e.g. ReLU) and do not have learnable weight matrices during message-passing.
>
> In our revised paper, we clarified it with the sentence  (Page 5, 2 lines from the top): *“Compared to general GNNs, the node embeddings of linear GNNs are given by closed-form formula.”*
>
> > W6. "It results in low value when node i and node j have heterophily embeddings, even if they are connected by an edge." This is not correct, see [2,3,6].
>
> The statement in the paper is correct, when we use **structure** embeddings (C1, Section 3.2, Page 3) in a (structure-)heterophily graph.
>
> In our revised paper, we clarified it with the sentence (Page 5, Section 4.1): *“However, even if node $i$ and node $j$ are connected by an edge, it may result in low value if they are expected to have dissimilar embeddings (e.g. structure embeddings in a bipartite graph).”*
>
> In more detail, in the cited papers [2, 3, 6], these are node features propagated through graph structure, which are similar to our derived **propagated** embeddings (C4 and C5).
> As discussed in W4, the **propagated** embeddings of connected nodes can be similar in a (label-)heterophily graph, while the **structure** embeddings will not be similar in a (structure-)heterophily graph.

---

> ### Author Response · Authors · 2023-11-20
> **Looking forward to your response!**
>
> Dear Reviewer ANgE,
>
> Thank you again for your thoughtful review.
> Based on your suggestions, we added the citations into the related works and clarified our wording.
> Please let us know your thoughts or if there is anything else we can do to address your comments.
>
> Best,
>
> Authors

---

> > ### Comment · Reviewer_ANgE · 2023-11-22
> >
> > Thanks for the response. The authors have addressed most of my concerns and I will raise my rating to 6. Remember to elaborate the statement and the proof of your theorem. Good luck.

---

> > > ### Author Response · Authors · 2023-11-22
> > > **Thank you for your response!**
> > >
> > > We truly appreciate your response and suggestion!

---

### Official Review · Reviewer_WXaL · 2023-11-01

**Soundness:** 3 good
**Presentation:** 3 good
**Contribution:** 3 good
**Rating:** 8
**Confidence:** 3

**Summary:**

The paper proposes a framework called NETINFOF for measuring and exploiting network usable information (NUI) in node-attributed graphs. The authors aim to determine if a graph neural network (GNN) will perform well on a given task by assessing the information present in the graph structure and node features. NETINFOF consists of two components: NETINFOF PROBE, which measures NUI without model training, and NETINFOF ACT, which solves link prediction and node classification tasks. The framework offers several advantages, including generality, principled approach, effectiveness, and scalability. The authors demonstrate the superiority of NETINFOF in identifying NUI and its performance in real-world datasets compared to general GNN baselines.

**Strengths:**

1. The paper introduces the novel NETINFOF framework, which addresses the problem of measuring and exploiting network usable information in graph for GNNs.
2. The paper demonstrates sound technical claims, supported by theoretical guarantees and empirical evaluations on synthetic and real-world datasets.
3. The writing style is clear, and the paper is well-organized, making it easy to understand the proposed framework and its contributions.
4. The paper's contributions are promising as they provide a practical tool (NETINFOF) for assessing the usefulness of graph structure and node features in GNN tasks. The framework shows promising results and outperforms existing baselines in link prediction.

**Weaknesses:**

1. How does the NETINFOF framework handle noisy or incomplete graph data? Can it effectively measure and exploit network usable information in such scenarios?

2. Are there specific types of graph structures or node features for which NETINFOF may not perform well? How robust is the framework in diverse graph settings?

3. I have concerns regarding the sensitivity of NETINFOF to the estimated compatibility matrix (H) used in the framework. It would be beneficial if the authors could provide additional empirical results that examine the performance of NETINFOF under different label rates, as label rates can significantly impact the correctness of H.

**Questions:**

see weaknesses

---

> ### Author Response · Authors · 2023-11-16
> **Response to Reviewer WXaL**
>
> We thank the reviewer for the positive and detailed comments, and give us valuable suggestions.
> We address the questions one by one.
>
> > W1. How does the NETINFOF framework handle noisy or incomplete graph data? Can it effectively measure and exploit network usable information in such scenarios?
>
> Yes, in such cases, NetInfoF will give low NetInfoF_Score to the corresponding noisy components in Section 3.2 (Page 3), and down-weigh them for the upcoming predictions.
>
> Different graph scenarios are studied in Table 6 in Appendix C.1 (Page 16):
> For example, if the given graph has useful structure but very noisy features, NetInfoF gives high scores to the structural embeddings (C1 and C2), and a low score to the feature embedding (C3).
>
> > W2. Are there specific types of graph structures or node features for which NETINFOF may not perform well? How robust is the framework in diverse graph settings?
>
> NetInfoF makes no assumption for either the graph structure or the node feature.
> In fact, as shown in Table 2 (Page 8) and Table 9 (Page 17), NetInfoF successfully tackles multiple graph scenarios on link prediction and node classification, respectively.
>
> > W3. I have concerns regarding the sensitivity of NETINFOF to the estimated compatibility matrix (H) used in the framework. It would be beneficial if the authors could provide additional empirical results that examine the performance of NETINFOF under different label rates.
>
> Thank you for your valuable suggestion.
> For link prediction, even a small fraction of existing links (=”labels”) is enough for estimating the compatibility matrix, as we show in the following table.
> The table shows two medium size datasets “Computers” and “Actor”, where we vary the ratio of positive edges used for estimating the compatibility matrix.
> NetInfoF still works well even with 10% of the edges for the compatibility matrix estimation, and it also performs better than without using the compatibility matrix.
>
> For large graphs, we do not expect NetInfoF to be sensitive to the ratio of edge labels, since it requires only a few edge labels to well estimate the compatibility matrix. For example, in “ogbn-Products”, there are 43M (= 62M * 70%) edges in the training set; as we mentioned in T3 in Section 4.1, we sample only 200K for estimating the compatibility matrix, which is 0.5% (= 200K / 43M) edges in the training set.
>
> | Edge Label Ratio | w/o CM | 10% | 30% | 50% | 70% |
> | -------- | -------- | -------- | -------- | -------- | -------- |
> | Comp.    | $27.9\pm0.2$ | $29.4\pm2.7$ | $33.0\pm1.8$ | $31.7\pm3.4$ | $31.1\pm1.9$ |
> | Actor    | $29.5\pm1.8$ | $33.3\pm1.1$ | $33.9\pm1.1$ | $34.3\pm0.6$ | $36.2\pm1.2$ |

---

> ### Author Response · Authors · 2023-11-20
> **Looking forward to your response!**
>
> Dear Reviewer WXaL,
>
> Thank you again for your thoughtful review.
> Based on your suggestions, we conduct an additional experiment on the sensitivity of NetInfoF on the edge label ratio.
> Please let us know your thoughts or if there is anything else we can do to address your comments.
>
> Best,
>
> Authors

---

### Official Review · Reviewer_96be · 2023-11-10

**Soundness:** 3 good
**Presentation:** 3 good
**Contribution:** 4 excellent
**Rating:** 8
**Confidence:** 4

**Summary:**

The paper presents a novel framework, NETINFOF, designed to quantify and leverage Network Usable Information (NUI) for graph tasks like link prediction and node classification. The approach tackles the need for extensive model training by using NETINFOF_PROBE to measure NUI directly from graph data, which is a significant improvement  from traditional GNNs that rely on trained low-dimensional representations. The NETINFOF_ACT component then uses this measured information to enhance the performance of graph tasks. The framework's robustness is tested on synthetic datasets designed to include various graph scenarios and validated on real-world datasets, showing superior results in link prediction tasks over standard GNN models. This paper's primary contribution is a method that can quickly assess a graph's usefulness for a task, offering a theoretical lower bound for accuracy without the computational overhead of model training​. The empirical study of this paper is very strong, surpassing most GNN methods both in accuracy and scalability.

**Strengths:**

- **Methodological Innovation**: NetInfoF introduces a novel approach to measure NUI directly from the graph structure and node features, which is a different from traditional methods that rely heavily on model training. This innovation could have a broad impact, particularly in scenarios where computational efficiency is paramount​. Also, the adjustments of node similarity using compatibility matrix w/ negative edges is very interesting.
- **Teoretical Foundation and Empirical Validation**: The paper provides a theoretical analysis for the NetInfoF_Score, presenting it as a lower bound for the accuracy of a GNN on a given task. This theoretical contribution is well-supported by empirical evidence on both synthetic datasets and real-life datasets.
- **Scalability**: The demonstrated scalability of NetInfoF, especially its linear scaling with input size (Fig. 6), and the use of significantly fewer parameters compared to GAE methods (1280 vs 279k) demonstrates its potential for application in large-scale graph tasks, presenting a substantial advancement in the practical deployment of GNNs​. Though NetInfoF is slower than SlimG [1], NetInfoF has better accuracy.

Combining 2nd and 3rd points, NetInfoF is better than GAE methods both in accuracy and scalability empirically, which is a significant contribution in link prediction tasks.






[1] Yoo, Jaemin, Meng-Chieh Lee, Shubhranshu Shekhar, and Christos Faloutsos. "Slendergnn: Accurate, robust, and interpretable gnn, and the reasons for its success." arXiv preprint arXiv:2210.04081 (2022).

**Weaknesses:**

1. The proof of Theorem 2 is problematic due to the incorrect use of conditional entropy. The proof incorrectly states the conditional entropy in Appendix A.2 as:
$$
H(Y|X) = \sum_{i=1}^{m} p_i \left(-\sum_{j=1}^{n} p_{ij} \log_2 p_{ij}\right),
$$
which is incorrect because it uses the joint probabilities $p_{ij}$ instead of the conditional probabilities. The correct expression for conditional entropy, which is based on the conditional probabilities, is:
$$
H(Y|X) = -\sum_{i=1}^{m} \sum_{j=1}^{n} p_{ij} \log_2 P(Y=y_j|X=x_i),
$$
where $P(Y=y_j|X=x_i) = \frac{p_{ij}}{p_i}$ is the conditional probability of $Y$ given $X=x_i$. And $p_i = \sum_{j=1}^{n} p_{ij}$ represents the marginal probability of $X$ taking the value $x_i$. This definition adheres to the fundamental property that the conditional probabilities for a given $x_i$ should sum to 1, i.e., $\sum_{j=1}^{n} P(Y=y_j|X=x_i) = 1$.

The misuse of conditional entropy in the proof leads to an erroneous application of Jensen's Inequality and subsequently invalidates the derived bound on the NetInfoF score.

2. (Minor) Given the paper is mostly dealing with link prediction tasks. It will be beneficial to include some subgraph-based methods as baselines, such as SEAL [1]. I understand that NetInfoF is intended as an advancement beyond traditional GNN approaches, but including subgraph-based methods will give the community a more comprehensive evaluation of NetInfoF's capabilities. It's evident NetInfoF is way more scalable than subgraph-based methods, but subgraph-based methods are still SOTA on some datasets, e.g. Ogbl-collab.







[1] Zhang, Muhan, and Yixin Chen. "Link prediction based on graph neural networks." Advances in neural information processing systems 31 (2018).

**Questions:**

1. Can the authors provide a more detailed justification for Theorem 2, particularly in light of the concerns highlighted in the first weakness?  I will raise my score if the authors can give a reasonable update.
2. How are the "Our bound" lines in Figures 3 and 4 derived? Additional details on this calculation would aid in understanding these figures.
3. How NetInfoF_Probe estimates and calculates the discretizer in section 4.2? Maybe some examples or explanations will help reader understand this.
4. An ablation study detailing the individual contributions of the five different node embedding components mentioned in Section 3.2 would be beneficial. Are these components equally influential in terms of accuracy, or do some weigh more significantly than others?

---

> ### Author Response · Authors · 2023-11-16
> **Response to Reviewer 96be (1/2)**
>
> We thank the reviewer for the positive and detailed comments, and truly appreciate that you even check the proof carefully.
> We address the questions one by one.
>
> > W1 & Q1. Can the authors provide a more detailed justification for Theorem 2, particularly in light of the concerns highlighted in the first weakness?
>
> Thank you for pointing out the problem, which is caused by the mistyping of the notations.
> We update the proof of Theorem 2 in Appendix A.2 (Page 13), where our conclusion remains the same.
>
> > Q2. How are the "Our bound" lines in Figures 3 and 4 derived?
>
> We update the annotations in Figure 3 (Page 4) to make it more clear.
> In more detail, according to Theorem 2, the training accuracy can not be lower than NetInfoF_Score, and thus our bound is the dash $x=y$ line in Figure 3.
> We show that even for validation accuracy, the results still obey Theorem 2.
> In Figure 4, the fitting lines have high $R^2$ values, indicating that NetInfoF_Score is highly correlated with the test performance.
> Since it is difficult to connect a metric to the test performance, the ideal metric is expected to be highly correlated with the test performance.
>
> > Q3. How NetInfoF_Probe estimates and calculates the discretizer in section 4.2?
>
> In Section 4.2, for link prediction, we did the quantile bucketization with $k=8$ bins on the adjusted node similarity.
> Due to space limitation, the detailed algorithm was provided in Algorithm 2 in the appendix (Page 15).
>
> Moreover, we conduct a sensitivity analysis on a medium size dataset “Computers" (see table below) and show that the result is insensitive to the exact value of $k$, forming a plateau when $k$ increases.
> We put these results in the appendix (Table 8 and Figure 7, Page 16) in our revised paper.
>
> | Number of Bins | k=4 | k=8 | k=16 | k=32 | k=64 |
> | -------- | -------- | -------- | -------- | -------- | -------- |
> | C1: U    | $74.2\pm0.3$ | $79.7\pm0.2$ | $80.4\pm0.3$ | $80.8\pm0.3$ | $81.0\pm0.3$ |
> | C2: R    | $76.5\pm0.2$ | $80.4\pm0.0$ | $81.2\pm0.3$ | $81.4\pm0.3$ | $81.5\pm0.3$ |
> | C3: F    | $63.8\pm0.1$ | $64.9\pm0.2$ | $65.0\pm0.1$ | $65.0\pm0.1$ | $65.1\pm0.1$ |
> | C4: P    | $77.3\pm0.2$ | $82.1\pm0.1$ | $83.0\pm0.3$ | $83.3\pm0.3$ | $83.4\pm0.3$ |
> | C5: S    | $77.3\pm0.1$ | $82.5\pm0.1$ | $83.3\pm0.1$ | $83.6\pm0.2$ | $83.7\pm0.2$ |
>
> > Q4. An ablation study detailing the individual contributions of the five different node embedding components mentioned in Section 3.2 would be beneficial. Are these components equally influential in terms of accuracy, or do some weigh more significantly than others?
>
> In short, different components have variable impact depending on the input graph.
> It is an advantage of NetInfoF that it can pinpoint which component is useful.
>
> In more detail, we studied the information of individual components in Appendix C.1 (Page 16) on the synthetic datasets.
> In addition, we further conduct this experiment on real-world datasets, please find additional results in the following table (major component: in bold).
>
> This conclusion is similar to the one we have in Appendix C.1 (Page 16), where different input graphs may have different preferences on the components depending on the underlying scenarios.
> Therefore, by combining all five components of node embeddings, NetInfoF is effective as well as robust.
> We put this table in the appendix (Table 7, Page 16) in our revised paper.
>
> | Dataset | Cora | CiteSeer | PubMed | Comp. | Photo | ArXiv | Products | Cham. | Squirrel | Actor | Twitch | Pokec |
> | -------- | -------- | -------- | -------- | -------- | -------- | -------- | -------- | -------- | -------- | -------- | -------- | -------- |
> | C1: U    | $48.3\pm0.8$ | $36.9\pm2.5$ | $43.7\pm1.8$ | $24.0\pm1.8$ | $38.5\pm2.6$ | $18.1\pm0.6$ | $13.2\pm0.3$ | $75.0\pm5.2$ | $12.3\pm1.8$ | $23.2\pm2.2$ | $\mathbf{15.8\pm0.2}$ | $\mathbf{16.2\pm0.6}$ |
> | C2: R    | $61.5\pm1.2$ | $50.0\pm1.9$ | $47.9\pm0.8$ | $19.4\pm0.8$ | $36.8\pm3.3$ | $12.3\pm1.0$ | $09.4\pm0.7$ | $64.6\pm3.2$ | $08.2\pm1.4$ | $\mathbf{34.2\pm2.3}$ | $01.5\pm0.2$ | $05.2\pm0.2$ |
> | C3: F    | $58.3\pm3.2$ | $71.5\pm2.4$ | $41.6\pm0.4$ | $07.1\pm0.4$ | $15.6\pm1.2$ | $04.9\pm0.2$ | $00.4\pm0.1$ | $10.7\pm1.0$ | $00.6\pm0.1$ | $02.6\pm0.5$ | $01.7\pm0.6$ | $00.5\pm0.2$ |
> | C4: P    | $67.3\pm1.8$ | $60.9\pm2.5$ | $48.1\pm1.8$ | $\mathbf{28.6\pm2.4}$ | $\mathbf{42.4\pm1.4}$ | $\mathbf{34.2\pm1.0}$ | $27.6\pm0.5$ | $\mathbf{84.3\pm1.8}$ | $\mathbf{20.9\pm2.3}$ | $13.1\pm1.1$ | $07.3\pm1.0$ | $12.4\pm0.8$ |
> | C5: S    | $\mathbf{82.4\pm1.0}$ | $\mathbf{88.7\pm1.5}$ | $\mathbf{63.3\pm1.1}$ | $29.0\pm2.3$ | $40.3\pm1.1$ | $33.6\pm1.1$ | $\mathbf{28.1\pm0.7}$ | $80.5\pm2.7$ | $19.2\pm1.2$ | $22.3\pm0.9$ | $02.7\pm1.8$ | $\mathbf{16.2\pm0.6}$ |

---

> ### Author Response · Authors · 2023-11-16
> **Response to Reviewer 96be (2/2)**
>
> > W2. It will be beneficial to include some subgraph-based methods as baselines, such as SEAL.
>
> We add the citations in Section 2 and run the experiments of SEAL with the default settings provided by OGB and set the hidden size to 128 as the same as other baselines, please find additional results in the following table.
>
> The experiments are run by an AWS EC2 g4dn.metal instance, with 384GB RAM and Tesla T4 GPUs.
> Noting that SEAL runs out of memory (O.O.M.) in the preprocessing step for several graphs, and exceeds time limit (T.L.E.) on “Photo”, taking more than 11 hours to run one (out of five) split.
> Furthermore, as shown by Table 1 (Page 2), subgraph GNNs miss several desired properties focused by this study, i.e., SEAL is not scalable and can not be used on node classification, and NetInfoF is the only one that matches all properties.
> Due to the time constraints, we are not able to conduct hyperparameter search and will conduct a more rigorous experiment in the next version.
>
> | Dataset | Cora | CiteSeer | PubMed | Comp. | Photo | ArXiv | Products | Cham. | Squirrel | Actor | Twitch | Pokec |
> | -------- | -------- | -------- | -------- | -------- | -------- | -------- | -------- | -------- | -------- | -------- | -------- | -------- |
> | SEAL     | $65.2\pm0.5$ | $60.0\pm1.1$ | $\mathbf{60.9\pm2.1}$ | $\color{red}{O.O.M.}$ | $\color{red}{T.L.E.}$ | $\color{red}{O.O.M.}$ | $\color{red}{O.O.M.}$ | $\mathbf{87.6\pm0.9}$ | $\color{red}{O.O.M.}$ | $\mathbf{43.8\pm0.8}$ | $\color{red}{O.O.M.}$ | $\color{red}{O.O.M.}$ |
> | NetInfoF | $\mathbf{81.3\pm0.6}$ | $\mathbf{87.3\pm1.3}$ | $59.7\pm1.1$ | $\mathbf{31.1\pm1.9}$ | $\mathbf{46.8\pm2.2}$ | $\mathbf{39.2\pm1.8}$ | $\mathbf{35.2\pm1.1}$ | $86.9\pm2.3$ | $\mathbf{24.2\pm2.0}$ | $36.2\pm1.2$ | $\mathbf{19.6\pm0.7}$ | $\mathbf{31.3\pm0.5}$ |

---

> ### Author Response · Authors · 2023-11-20
> **Looking forward to your response!**
>
> Dear Reviewer 96be,
>
> Thank you again for your thoughtful review.
> Based on your suggestions, we corrected the proof of Theorem 2 and conducted an additional experiment on SEAL.
> Please let us know your thoughts or if there is anything else we can do to address your comments.
>
> Best,
>
> Authors

---

> > ### Comment · Reviewer_96be · 2023-11-21
> > **Response to Authors**
> >
> > Thank the authors for their detailed reply and experiments and sorry for my late reply. In general, I think the authors deal with my concern effectively. I have raised my score to 8 for now. And I am looking forward to discussing with other reviewers about the paper to see if I have missed any points.
> >
> > W1 & Q1: Proof of theorem looks good to me now.
> >
> > Q2: Thanks for the clarification.
> >
> > Q3: It is interesting to see that NetInfoF_Probe is insensitive to $k$.
> >
> > Q4: The ablation study is great and reveals interesting patterns. Can the authors provide some insight about why certain components perform better on homophily/heterophily datasets?
> >
> > W2: It’s great to see NetInfoF outperform SEAL on certain datasets and also turns out to be scalable on large datasets. It’s a great supplement to this work.
> >
> > In general, I am satisfied with the current paper. And it seems that the authors have made significant efforts on responding to reviewers’ feedback.

---

> > > ### Author Response · Authors · 2023-11-21
> > > **Thank you for your response!**
> > >
> > > We truly appreciate your response and are grateful that you noticed our effort on responding!
> > >
> > > > Q4: …Can the authors provide some insight about why certain components perform better on homophily/heterophily datasets?
> > >
> > > Thank you for your valuable suggestion, we plan to add the following explanations:
> > >
> > > *“For all homophily graphs (on the left of the vertical line in Table 7), the propagated embeddings (C4 and C5) are useful for link prediction;
> > > for non-homophily graphs (on the right of the vertical line in Table 7), it depends on specifics of the dataset.”*

---

### Author Response · Authors · 2023-11-16
**Common Response**

We thank all reviewers for the insightful and constructive comments on our work.
We have carefully addressed the comments and incorporated the changes into our revised paper, highlighted with the red font.

---

### Meta-Review · Area_Chair_g7DB · 2023-12-15

**Metareview:**

This paper investigates an interesting problem, network usable information (NUI), which aims to measure the extent to which the graph structure and node features carry useful information for a link prediction or node classification task. The proposed method seems technically sound, and could be practically very useful. There were questions and concerns regarding the proofs and clarity. But the authors have successfully addressed most of them and all reviewers ended up being positive about this paper.

**Justification For Why Not Higher Score:**

Overall, there is room to improve the writing and clarity of this paper.

**Justification For Why Not Lower Score:**

The proposed method is shown to be effective and could be practically very useful.

---

### Decision · Program_Chairs · 2024-01-16

Accept (spotlight)